# DEEPER-GXX: DEEPENING ARBITRARY GNNS

## ABSTRACT

Recently, motivated by real applications, a major research direction in graph neural networks (GNNs) is to explore deeper structures. For instance, the graph connectivity is not always consistent with the label distribution (e.g., the closest neighbors of some nodes are not from the same category). In this case, GNNs need to stack more layers, in order to find the same categorical neighbors in a longer path for capturing the class-discriminative information. However, two major problems hinder the deeper GNNs to obtain satisfactory performance, i.e., *vanishing gradient* and *over-smoothing*. On one hand, stacking layers makes the neural network hard to train as the gradients of the first few layers vanish. Moreover, when simply addressing vanishing gradient in GNNs, we discover the *shading neighbors* effect (i.e., stacking layers inappropriately distorts the non-IID information of graphs and degrade the performance of GNNs). On the other hand, deeper GNNs aggregate much more information from common neighbors such that individual node representations share more overlapping features, which makes the final output representations not discriminative (i.e., overly smoothed). In this paper, for the first time, we address both problems to enable deeper GNNs, and propose **Deeper-GXX**, which consists of the Weight-Decaying Graph Residual Connection module (WDG-ResNet) and Topology-Guided Graph Contrastive Loss (TGCL). Extensive experiments on real-world data sets demonstrate that Deeper-GXX outperforms state-of-the-art deeper baselines.

## 1 INTRODUCTION

Graph neural networks (GNNs) have been proven successful at modeling graph data by extracting node hidden representations that are effective for many downstream tasks. In general, they are realized by the message passing schema and aggregate neighbor features to obtain node hidden representations (Kipf & Welling, 2017; Hamilton et al., 2017a; Velickovic et al., 2018; Xu et al., 2019). Recently, the surge of big data makes graphs' structural and attribute information much more complex and uncertain, which urges the researchers to make GNNs deeper (i.e., stacking more graph neural layers), in order to capture more meaningful information for better performance. For example, in social media, people from different categories (e.g., occupation, interests, etc.) are often connected (e.g., become friends), and users' immediate neighbor information may not reflect their categorical information. Thus, deepening GNNs is necessary to identify the neighbors from the same category in a longer path (e.g., $k$-hop neighbors), and to aggregate their features to obtain the class-discriminative node representations. To demonstrate the benefit of deeper GNNs, we conduct a case study shown in Figure 1 (See the detailed experimental setup in Appendix A.1). In Figure 1a, we observe that the query node (the diamond in the black dashed circle) cannot rely on its closest labeled neighbor (the red star in the circle) to correctly predict its label (the blue). Only by exploring longer paths consisting of more similar neighbors are we able to predict its label as blue. Figure 1b compares the classification accuracy of shallow GNNs and deeper GNNs. We can see that deeper GNNs significantly outperform shallow ones by more than 11%, due to their abilities to explore longer paths on the graph. Similar observations of the benefits of deeper GNNs are also found in the missing feature scenario presented in Section 3.3.

However, simply stacking layers of GNNs can be problematic, due to *vanishing gradient* and *over-smoothing* issues. On one hand, increasing the number of neural layers can induce the hard-to-train model, where both the training error and test error are higher than shallow ones. This is mainly caused by the *vanishing gradient* issue (He et al., 2016), where the gradient of the first few layers vanish

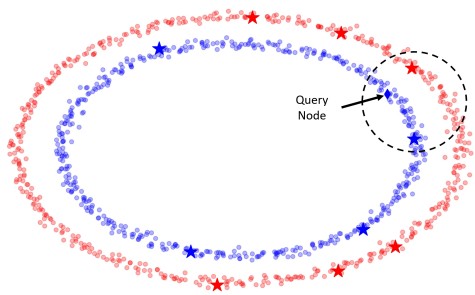 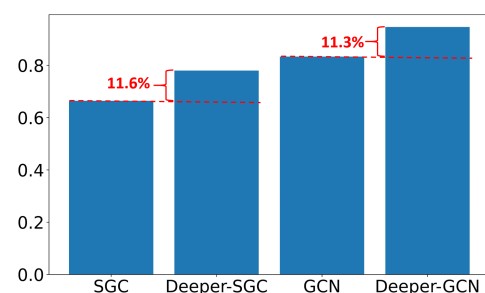

(a) Two groups of nodes in the semi-supervised setting. Stars are labeled, dots are unlabeled, and the diamond is the query node. Euclidean distance between two nodes indicates the edge connection.

(b) Comparison of node classification accuracy between shallow and deeper GNN models using data on the left. The deeper GNNs are realized by our Deeper-GXX with corresponding backbones.

Figure 1: A toy example to demonstrate the benefit of deeper GNN models.

such that the training loss could not be successfully propagated through deeper models. ResNet (He et al., 2016) has been proposed to address this issue. However, we discover that simply combining ResNet with GNNs still leads to the sub-optimal solution: as ResNet stacks layers, the importance of close neighbors' features gradually decreases during the GNN information aggregation process, and the faraway neighbor information becomes dominant. We call this effect as *shading neighbors*. On the other hand, GNN utilizes the message passing schema to aggregate neighbor features, in order to get class-discriminative node representations. However, by stacking more layers, each node begins to share more and more overlapping neighbor information during the aggregation process and the node representations gradually become indistinguishable (Li et al., 2018; Oono & Suzuki, 2020). This has been referred to as the *over-smoothing* issue, and it significantly affects the performance of downstream tasks such as node classification and link prediction.

In this paper, we study how to effectively stack GNN layers by addressing *shading neighbors* and *over-smoothing* at the same time. First, to address the *shading neighbors* caused by the direct application of ResNet on GNNs, we propose **Weight-Decaying Graph Residual Connection** (WDG-ResNet), which learns the weight of each residual connection layer (instead of setting it as 1 in ResNet), and further introduces a decaying factor to refine the weight of each layer. Interestingly, we find that the hyperparameter $\lambda$ of the weight decaying factor actually controls the number of effective layers in deeper GNNs based on the input graph inherent property, which is verified in Appendix A.5. Second, for addressing *over-smoothing*, we propose **Topology-Guided Graph Contrastive Loss** (TGCL) in the contrastive learning manner (van den Oord et al., 2018), where the hidden representations of the positive pairs should be closer, and those of the negative pairs should be pushed apart. Through theoretical and empirical analysis, we find that TGCL can be effectively and efficiently realized by only considering 1-hop neighbors as the positive pair and all the rest as negative pairs. Combining the proposed WDG-ResNet and TGCL, we propose an end-to-end model called **Deeper-GXX** to help arbitrary GNNs go deeper. Our contributions can be summarized as follows.

- We propose Weight-Decaying Graph Residual Connection (WDG-ResNet) to address the *shading neighbors* effect caused by vanilla ResNet in dealing with the *vanishing gradient* of GNNs.
- We propose Topology-Guided Graph Contrastive Loss (TGCL) to address the *over-smoothing* problem by encoding the graph topological information to the discriminative node representations.
- We combine the proposed WDG-ResNet and TGCL into an end-to-end model called **Deeper-GXX**, which is model-agnostic and can help arbitrary GNNs go deeper.
- Extensive experiments show that Deeper-GXX outperforms state-of-the-art deeper baselines.

## 2 PROPOSED METHOD

In this section, we begin with the overview of Deeper-GXX. Then, we provide the details of the proposed Weight-Decaying Graph Residual Connection (WDG-ResNet) and Topology-Guided Graph Contrastive Loss (TGCL), which address *shading neighbors* and *over-smoothing* problems, respectively. We formalize the graph embedding problem in the context of undirected graph $\mathcal{G} = (V, E, \boldsymbol{X})$, where $V$ consists of $n$ vertices, $E$ consists of $m$ edges, $\boldsymbol{X} \in \mathbb{R}^{n \times d}$ denotes the feature

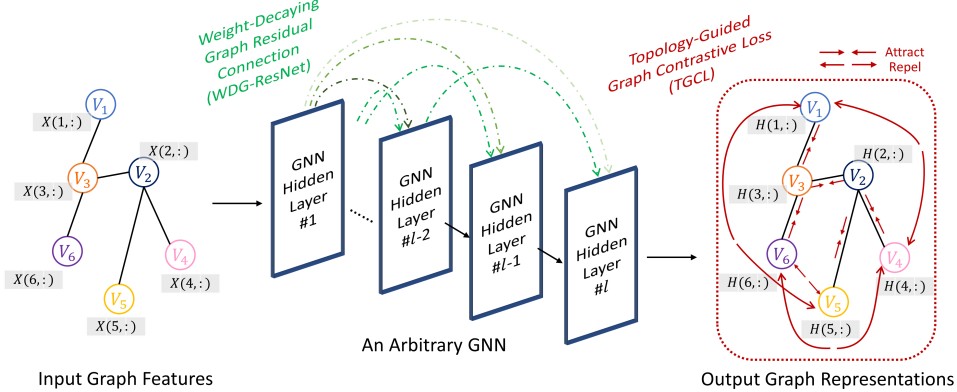

Figure 2: An arbitrary GNN with the proposed Deeper-GXX.

matrix and $d$ is the feature dimension. We let $\boldsymbol{A} \in \mathbb{R}^{n \times n}$ denote the adjacency matrix and denote $\boldsymbol{A}_i \in \mathbb{R}^n$ as the adjacency vector for node $v_i$. $\boldsymbol{H}_i \in \mathbb{R}^h$ is the hidden representation vector of $v_i$.

## 2.1 OVERVIEW OF DEEPER-GXX

The overview of our proposed Deeper-GXX is shown in Figure 2. The green dash line stands for Weight-Decaying Graph Residual Connection (WDG-ResNet). In WDG-ResNet, after the current hidden representation $\boldsymbol{H}^{(l)}$ is generated by the $l$-th layer of arbitrary GNNs, $\boldsymbol{H}^{(l)}$ will be adjusted by its second last layer $\boldsymbol{H}^{(l-2)}$ and the first layer $\boldsymbol{H}^{(1)}$ with proper weights. The red dash line stands for Topology-Guided Graph Contrastive Loss (TGCL). In TGCL, we first need to sample positive node pairs and negative node pairs based on the input graph topology such that the hidden representations of positive node pairs get closer and negative ones are pushed farther apart. After introducing the TGCL loss to GNNs, the overall loss function $\mathcal{L}_{overall}$ of Deeper-GXX is expressed as follows.

$$\mathcal{L}_{overall} = \mathcal{L}_{GNN} + \alpha \mathcal{L}_{\text{TGCL}} \tag{1}$$

where $\mathcal{L}_{GNN}$ denotes the loss of the downstream task using an arbitrary GNN model (e.g., node classification in GCN), $\mathcal{L}_{\text{TGCL}}$ is the TGCL loss, and $\alpha$ is a constant hyperparameter. Deeper-GXX combines WDG-ResNet and TGCL to address *shading neighbors* and *over-smoothing* problems. The design of WDG-ResNet and TGCL are introduced in the following subsections.

## 2.2 WEIGHT-DECAYING GRAPH RESIDUAL CONNECTION (WDG-RESNET)

As we increase the number of layers, one unavoidable problem brought by neural networks is the *vanishing gradient*, which means that the first several layers of the deeper neural network become hard to optimize as their gradients vanish during the training process (He et al., 2016). Currently, nascent deeper GNN methods (Zhao & Akoglu, 2020; Rong et al., 2020; Li et al., 2019) solve this problem by adding ResNet (He et al., 2016) on graph neural networks. Taking GCN (Kipf & Welling, 2017) as an example, the graph residual connection is expressed as follows.

$$\boldsymbol{H}^{(l)} = \sigma(\hat{\boldsymbol{A}} \boldsymbol{H}^{(l-1)} \boldsymbol{W}^{(l-1)}) + \boldsymbol{H}^{(l-2)} \tag{2}$$

where $l(\geq 2)$ denotes the index of layers, $\boldsymbol{H}^{(l-1)}$ and $\boldsymbol{H}^{(l-2)}$ are the hidden representations, $\sigma(\cdot)$ is the activation function, $\boldsymbol{W}^{(l-1)}$ is the learnable weight matrix, and $\hat{\boldsymbol{A}}$ is the re-normalized self-looped adjacency matrix with $\hat{\boldsymbol{A}} = \tilde{\boldsymbol{D}}^{-\frac{1}{2}} \tilde{\boldsymbol{A}} \tilde{\boldsymbol{D}}^{-\frac{1}{2}}$ and $\tilde{\boldsymbol{A}} = \boldsymbol{A} + \boldsymbol{I}$. In ResNet, the residual connection connects the current layer and its second last layer. Without loss of generality, we assume the last layer of GNNs is stacked by ResNet, i.e., $l$ is divisible by 2. Then, by extending $\boldsymbol{H}^{(l-2)}$ iteratively (i.e., substituting it with its previous residual blocks), the above Eq. 2 could be rewritten as follows.

$$\boldsymbol{H}^{(l)} = \sigma(\hat{\boldsymbol{A}} \boldsymbol{H}^{(l-1)} \boldsymbol{W}^{(l-1)}) + \boldsymbol{H}^{(l-2)}$$

$$\boldsymbol{H}^{(l)} = \sigma(\hat{\boldsymbol{A}} \boldsymbol{H}^{(l-1)} \boldsymbol{W}^{(l-1)}) + \sigma(\hat{\boldsymbol{A}} \boldsymbol{H}^{(l-3)} \boldsymbol{W}^{(l-3)}) + \boldsymbol{H}^{(l-4)}$$

$$= \underbrace{\sigma(\hat{\boldsymbol{A}} \boldsymbol{H}^{(l-1)} \boldsymbol{W}^{(l-1)}) + \sigma(\hat{\boldsymbol{A}} \boldsymbol{H}^{(l-3)} \boldsymbol{W}^{(l-3)}) + \cdots}_{\text{Information aggregated from the faraway neighbors}} + \underbrace{\sigma(\hat{\boldsymbol{A}} \boldsymbol{H}^{(i)} \boldsymbol{W}^{(i)}) + \cdots + \sigma(\hat{\boldsymbol{A}} \boldsymbol{H}^{(1)} \boldsymbol{W}^{(1)})}_{\text{Information aggregated from the nearest neighbors}}$$

$$\tag{3}$$

According to the GNN theoretical analysis, stacking $l$ layers and getting $\boldsymbol{H}^{(l)}$ in GNNs can be interpreted as aggregating $l$-hop neighbors' feature information for the node hidden representations (Xu et al., 2019). As shown in Eq. 3, when we stack more layers in GNNs, the information collected from faraway neighbors becomes dominant (as there are more terms regarding the information from faraway neighbors), compared with the information collected from the nearest neighbors (e.g., 1-hop or 2-hop neighbors). This phenomenon contradicts the general intuition that the close neighbors of a node carry the most important information, and the importance degrades with faraway neighbors. Formally, we describe this phenomenon as *shading neighbors* effect when stacking graph neural layers, as the importance of the nearest neighbors is diminishing. We show that *shading neighbors* effect downgrades the GNNs performance in downstream tasks in Section 3.4.

To address the shading neighbors effect, we propose Weight-Decaying Graph Residual Connection (WDG-ResNet). Here, we first introduce the formulation and then provide the insights regarding why it can address the problem. Specifically, WDG-ResNet introduces the layer similarity and weight decaying factor as follows.

$$
\begin{aligned}
\tilde{\boldsymbol{H}}^{(l)} &= \sigma(\hat{\boldsymbol{A}}\boldsymbol{H}^{(l-1)}\boldsymbol{W}^{(l-1)}), \quad \textit{/*l-th layer of an arbitrary GNN*/} \\
\boldsymbol{H}^{(l)} &= sim(\boldsymbol{H}^{(1)}, \tilde{\boldsymbol{H}}^{(l)}) \cdot e^{-l/\lambda} \cdot \tilde{\boldsymbol{H}}^{(l)} + \boldsymbol{H}^{(l-2)}, \quad \textit{/*residual connection*/} \\
&= e^{cos(\boldsymbol{H}^{(1)}, \tilde{\boldsymbol{H}}^{(l)}) - l/\lambda} \cdot \tilde{\boldsymbol{H}}^{(l)} + \boldsymbol{H}^{(l-2)}
\end{aligned}
\tag{4}
$$

where $cos(\boldsymbol{H}^{(1)}, \tilde{\boldsymbol{H}}^{(l)}) = \frac{1}{n}\sum_i \frac{\boldsymbol{H}_i^{(1)}(\tilde{\boldsymbol{H}}_i^{(l)})^\top}{\|\boldsymbol{H}_i^{(1)}\|\|\tilde{\boldsymbol{H}}_i^{(l)}\|}$ measures the similarity between the $l$-th layer and the 1-st layer, and we use the exponential function to map the cosine similarity ranging from $\{-1, 1\}$ to $\{e^{-1}, e^1\}$, to avoid the negative similarity weights. The term $e^{-l/\lambda}$ is the decaying factor to further adjust the similarity weight of $\tilde{\boldsymbol{H}}^{(l)}$, where $\lambda$ is a constant hyperparameter.

Different from ResNet (He et al., 2016), we add the learnable similarity $sim(\boldsymbol{H}^{(1)}, \tilde{\boldsymbol{H}}^{(l)})$ to essentially expand the hypothesis space of deeper GNNs. Additional to that, simply adding vanilla ResNet on GNNs will cause the shading neighbors effect. The introduced decaying factor $e^{-l/\lambda}$ can alleviate this negative effect, because it brings the layer-wise dependency to stacking operations, in order to preserve the graph hierarchical information when GNNs go deeper. Since $\lambda$ is a constant, the value of $e^{-l/\lambda}$ is decreasing while $l$ increases. Thus, the later stacked layer is always less important than previously stacked ones by the decaying weight, which addresses the shading neighbors effect. Without the decaying factor, the layer-wise weights are independent, and the shading neighbor effect still exists. Moreover, we visualize the layer-wise weight distribution of different residual connection methods (including our WDG-ResNet) and their effectiveness in addressing shading neighbors effect in Appendix A.4.

From another perspective, the hyperparameter $\lambda$ of the decaying factor actually controls the number of effective neural layers in deeper GNNs. For example, when $\lambda = 10$, the decaying factor for the 10-th layer is 0.3679 (i.e., $e^{-1}$); but for the 30-th layer, it is 0.0049 (i.e., $e^{-3}$). This decay limits the effective information aggregation scope of deeper GNNs, because the later stacked layers will become significantly less important. Based on this controlling property of $\lambda$, a natural follow-up question is whether its value depends on the property of input graphs. Interestingly, through our experiments, we find that *the optimal $\lambda$ is very close to the diameter of input graphs (if it is connected) or the largest component (if it does not have many separate components)*. This observation verifies our conjecture regarding the property of $\lambda$ (i.e., it controls the number of effective layers or number of hops during the message passing aggregation schema of GNNs). Hence, the value of $\lambda$ can be searched around the diameter of the input graph, and we discuss the details in the Appendix A.5.

**Simplified WDG-ResNet**. In the experiments, we observe that if the number of layers of GNNs is too large (*e.g.*, 50 layers or more), the computational cost of adding the similarity function $sim(\boldsymbol{H}^{(1)}, \tilde{\boldsymbol{H}}^{(l)})$ at each graph residual connection layer can be expensive. To accelerate the computation, we formulate a simpler version of Eq. 4 by removing the $sim(\boldsymbol{H}^{(1)}, \tilde{\boldsymbol{H}}^{(l)})$ but keeping the decaying factor. The simplified version of WDG-ResNet is expressed as follows.

$$
\begin{aligned}
\boldsymbol{H}^{(l)} &= e^{-l/\lambda}\sigma(\boldsymbol{A}\boldsymbol{H}^{(l-1)}\boldsymbol{W}^{(l-1)}) + \boldsymbol{H}^{(l-2)} \\
&= e^{-l/\lambda}\sigma(\boldsymbol{A}\boldsymbol{H}^{(l-1)}\boldsymbol{W}^{(l-1)}) + e^{-(l-2)/\lambda}\sigma(\boldsymbol{A}\boldsymbol{H}^{(l-3)}\boldsymbol{W}^{(l-3)}) + e^{-(l-4)/\lambda}\sigma(\boldsymbol{A}\boldsymbol{H}^{(l-5)}\boldsymbol{W}^{(l-5)}) \\
&\quad + \cdots + e^{-4/\lambda}\sigma(\boldsymbol{A}\boldsymbol{H}^{(3)}\boldsymbol{W}^{(3)}) + e^{-2/\lambda}\sigma(\boldsymbol{A}\boldsymbol{H}^{(1)}\boldsymbol{W}^{(1)})
\end{aligned}
\tag{5}
$$

Compared with Eq. 4, Eq. 5 gets rid of the exponential cosine similarity measurement, which greatly reduces the computational cost. However, the simplified WDG-ResNet still keeps the decaying factor for the layer-wise dependency, such that the shading neighbors effect can still be alleviated.

### 2.3 TOPOLOGY-GUIDED GRAPH CONTRASTIVE LOSS (TGCL)

To address *over-smoothing*, current deeper GNN solutions depend on certain hard-to-acquire prior knowledge (e.g., important hyperparameters or sampling randomness) to get the discriminative node representations (Zhao & Akoglu, 2020; Rong et al., 2020; Zhou et al., 2020). Inspired by these solutions, we are seeking for an effective discriminative indicator that can be easily obtained from the input graph without any prior knowledge. Thus, we propose a novel contrastive regularization term, named Topology-Guided Graph Contrastive Loss (TGCL), to transfer this hard-to-acquire knowledge into the topology information of the input graph as follows.

$$\mathcal{L}_{\text{TGCL}} = - \mathbb{E}_{v_i \sim V} \, \mathbb{E}_{v_j \in \mathcal{N}_i} [\log \frac{\sigma_{ij} f(\boldsymbol{z}_i, \boldsymbol{z}_j)}{\sigma_{ij} f(\boldsymbol{z}_i, \boldsymbol{z}_j) + \sum_{v_k \in \bar{\mathcal{N}}_i} \gamma_{ik} f(\boldsymbol{z}_i, \boldsymbol{z}_k)}] \tag{6}$$
$$\sigma_{ij} = 1 - dist(\boldsymbol{A}_i, \boldsymbol{A}_j)/n, \ \ \gamma_{ik} = 1 + dist(\boldsymbol{A}_i, \boldsymbol{A}_k)/n$$

where $\boldsymbol{z}_i = g(\boldsymbol{H}_i^{(l)})$, $g(\cdot)$ is an encoder mapping $\boldsymbol{H}_i^{(l)}$ to another latent space, $f(\cdot)$ is a similarity function (e.g., $f(a, b) = \exp(\frac{ab^\top}{||a||||b||})$), $dist(\cdot)$ is a distance measurement function (e.g., hamming distance (Norouzi et al., 2012)), $\mathcal{N}_i$ is the set containing one-hop neighbors of node $v_i$, and $\bar{\mathcal{N}}_i$ is the complement of the set $\mathcal{N}_i$.

In Eq. 6, directly connected nodes $(v_i, v_j)$ form the positive pair, while not directly connected nodes $(v_i, v_k)$ form the negative pair. The intuition of this equation is to maximize the similarity of the representations of the positive pairs, and to minimize the similarity of the representations of the negative pairs, such that the node representations become discriminative. In the graph contrastive learning setting, designing the positive and negative pairs is very important, because it needs to effectively address the over-smoothing issue, and it should also be efficiently obtained given the topology of the input graph. We provide further discussion and theoretical analysis to demonstrate the effectiveness of our positive/negative pair sampling strategy for Eq. 6 in Appendix B.2.

With Eq. 6, we can analyze the proposed TGCL loss bound in terms of mutual information, as stated in Proposition 1. In particular, Proposition 1 demonstrates that the TGCL loss for the graph is the lower bound of the mutual information between two representations of a neighbor node pair. Therefore, minimizing TGCL is equivalent to maximizing the mutual information of connected nodes by taking graph topology information into consideration.

**Proposition 1** *Given a neighbor node pair sampled from the graph $\mathcal{G} = (V, E, \boldsymbol{X})$, i.e., nodes $v_i$ and $v_j$, we have $I(\boldsymbol{z}_i, \boldsymbol{z}_j) \geq -\mathcal{L}_{\text{TGCL}} + \mathbb{E}_{v_i \sim V} \log(|\mathcal{N}_i|)$, where $I(\boldsymbol{z}_i, \boldsymbol{z}_j)$ is the mutual information between two representations of the node pair $v_i$ and $v_j$, and $\mathcal{L}_{\text{TGCL}}$ is the topology-guided contrastive loss weighted by hamming distance measurement.*

The proof of this proposition can be found in Appendix B.1.

## 3 EXPERIMENTS

In this section, we demonstrate the performance of our proposed Deeper-GXX in terms of effectiveness by comparing it with state-of-the-art deeper GNN methods and self-ablations. In addition, we conduct a case study to show how the increasing number of layers influences the performance of deeper GNNs when the input graph has missing features.

### 3.1 EXPERIMENT SETUP

**Data sets:** The Cora (Lu & Getoor, 2003) data set is a citation network consisting of 5,429 edges and 2,708 scientific publications from 7 classes. The edge in the graph represents the citation of one paper by another. CiteSeer (Lu & Getoor, 2003) data set consists of 3,327 scientific publications which could be categorized into 6 classes, and this citation network has 9,228 edges. PubMed (Namata

et al., 2012) is a citation network consisting of 88,651 edges and 19,717 scientific publications from 3 classes. The Reddit (Hamilton et al., 2017b) data set is extracted from Reddit posts, which consists of 4,584 nodes and 19,460 edges. OGB-Arxiv (Wang et al., 2020) is a citation network, which consists of 1,166,243 edges and 169,343 nodes from 40 classes.

**Baselines:** We compare the performance of our method with the following baselines including one vanilla GNN model and four state-of-the-art deeper GNN models: (1) GCN (Kipf & Welling, 2017): the vanilla graph convolutional network; (2) GCNII (Chen et al., 2020b): an extension of GCN with skip connections and additional identity matrices; (3) DGN (Zhou et al., 2020): the differentiable group normalization for GNNs to normalize nodes within the same group and separate nodes among different groups; (4) PairNorm (Zhao & Akoglu, 2020): a GNN normalization layer designed to prevent node representations from becoming too similar; (5) DropEdge (Rong et al., 2020): a GNN-agnostic framework that randomly removes a certain number of edges from the input graph at each training epoch; (6) Deeper-GXX-S: using the simplified WDG-ResNet in Deeper-GXX.

**Configurations**: In the experiments, we follow the splitting strategy used in (Zhao & Akoglu, 2020) by randomly sampling 3% of the nodes as the training samples, 10% of the nodes as the validation samples, and the remaining 87% as the test samples. We set the learning rate to be 0.001 and the optimizer is RMSProp, which is one variant of ADAGRAD (Duchi et al., 2011). For fair comparison, we set the feature dimension of the hidden layer to be 50, the dropout rate to be 0.5, the weight decay rate to be 0.0005, and the total number of iterations to be 1500 for all methods. For Deeper-GXX and Deeper-GXX-S, we sample 10 instances and 5 neighbors for each class from the training set, $dist(\cdot)$ is the hamming distance, and $f(\cdot)$ is the cosine similarity measurement. The experiments are repeated 10 times if not otherwise specified. The code of our algorithm can be found in an anonymous link [*]. All of the real-world data sets are publicly available. The experiments are performed on a Windows machine with a 16GB RTX 5000 GPU.

**The detailed hyperparameters (e.g., $\lambda$ and $\alpha$) setting for the experimental results in each table can be found in Appendix A.2. Hyperparameter study and efficiency analysis could also be found in Appendix A.5 and A.6, respectively.**

## 3.2 EXPERIMENTAL ANALYSIS

In this subsection, we evaluate the effectiveness of the proposed method on four benchmark data sets by comparing it with state-of-the-art methods. The base model for all methods we used in these experiments is graph convolutional neural network (GCN) (Kipf & Welling, 2017). For fair comparison, we set the numbers of hidden layers to be 60 for all methods and the dimension of the hidden layer to be 50. The experiments are repeated 5 times and we record the mean accuracy as well as the standard deviation in Table 1.

Table 1: Accuracy of node classification on four benchmark data sets with 60 hidden layers. GCN is used as the backbone for all methods.

| Method | Cora | CiteSeer | PubMed | Reddit |
|---|---|---|---|---|
| GCN | $0.2962 \pm 0.0084$ | $0.2125 \pm 0.0140$ | $0.4172 \pm 0.0098$ | $0.1140 \pm 0.0136$ |
| PairNorm | $0.6759 \pm 0.0171$ | $0.4817 \pm 0.0197$ | $0.7883 \pm 0.0101$ | $0.9017 \pm 0.0241$ |
| DropEdge | $0.2911 \pm 0.0122$ | $0.2147 \pm 0.0184$ | $0.4162 \pm 0.0208$ | $0.1019 \pm 0.0324$ |
| GCNII | $0.6076 \pm 0.0050$ | $0.5775 \pm 0.0027$ | $0.8188 \pm 0.0030$ | $0.6969 \pm 0.0064$ |
| DGN | $0.7022 \pm 0.0079$ | $0.4398 \pm 0.0118$ | $0.7843 \pm 0.0032$ | $0.5122 \pm 0.0143$ |
| PairNorm + ResNet | $0.7394 \pm 0.0271$ | $0.5544 \pm 0.0166$ | $0.7985 \pm 0.0068$ | $0.9385 \pm 0.0102$ |
| DropEdge + ResNet | $0.1696 \pm 0.0271$ | $0.1951 \pm 0.0382$ | $0.5798 \pm 0.1501$ | $0.0950 \pm 0.0129$ |
| GCNII + ResNet | $0.7024 \pm 0.0075$ | $0.6051 \pm 0.0062$ | $0.8093 \pm 0.0047$ | $0.7538 \pm 0.0095$ |
| DGN + ResNet | $0.1543 \pm 0.0004$ | $0.2104 \pm 0.0000$ | $0.2086 \pm 0.0000$ | $0.1118 \pm 0.0000$ |
| Deeper-GXX-S | $0.8023 \pm 0.0117$ | $0.6544 \pm 0.0099$ | $\mathbf{0.8198 \pm 0.0012}$ | $0.9693 \pm 0.0036$ |
| Deeper-GXX | $\mathbf{0.8059 \pm 0.0028}$ | $\mathbf{0.6655 \pm 0.0117}$ | $0.8185 \pm 0.0016$ | $\mathbf{0.9721 \pm 0.0011}$ |

In the first five rows of Table 1, we observe that when we set 60 hidden layers as the reference, DropEdge has almost identical performance as the vanilla GCN. PairNorm, GCNII, and DGN increase the performance by more than 30% in four data sets compared with GCN. The latent reason is that

---

[*]`https://drive.google.com/file/d/1cbNI74lhTb3LsOKhgVHT1btNz20ZLb60/`
`view?usp=sharing`

the over-smoothing problem is alleviated to some extent, since these three are deliberately proposed to deal with over-smoothing problem in deeper GNNs. Besides, our proposed method (i.e., Deeper-GXX) and its simpler version (i.e., Deeper-GXX-S) outperform all of these baselines over four data sets. Addition to addressing the over-smoothing problem, another part of our outperformance can be credited to our proposed residual connection. To verify this conjecture, we further incorporate ResNet into PairNorm, DropEdge, DGN, GCNII, and DropEdge. Then, in the sixth to the eighth rows of Table 1, we can observe (1) some baselines like PairNorm and GCNII suffer from the vanishing gradient (e.g., the performance of PairNorm increases by 6% on Cora and 7% on CiteSeer, while the performance of GCNII rises to 75.38% on Reddit and 70.24% on Cora). Also, although GCNII designs a ResNet-like architecture, adding ResNet to GCNII can still boost its performance on three data sets; (2) Compared with their "+ResNet" versions, our Deeper-GXX and Deeper-GXX-S still outperform, which implies that our designed residual connection indeed contributes to the outperformance, and we do the ablation study to quantify each component's contribution of Deeper-GXX in Section 3.4; (3) Not all deeper baselines need ResNet. For example, DropEdge+ResNet almost maintains the performance, and DGN+ResNet drops the performance.

In addition to four small data sets, we also examine the the node classification performance of Deeper-GXX on a large-scale graph called OGB-Arxiv shown in Figure 3a and Figure 3b. In this experiment, we fix the feature dimension of the hidden layer as 100, the total iteration is set as 3000 and GCN is chosen as the base model. Due to the memory limitation, we set the number of layers as 10 for all baselines methods in Figure 3b for fair comparison. By observation, we find that (1) in Figure 3a, the performance of Deeper-GXX increases as we increase the number of layers, which verifies our conjuncture that increasing the number of layers indeed leads to better performance in large graphs due to more information aggregated from neighbors; (2) comparing with PairNorm, Deeper-GXX further boosts the performance by more than 5.6% on OGB-Arxiv data set in Figure 3b.

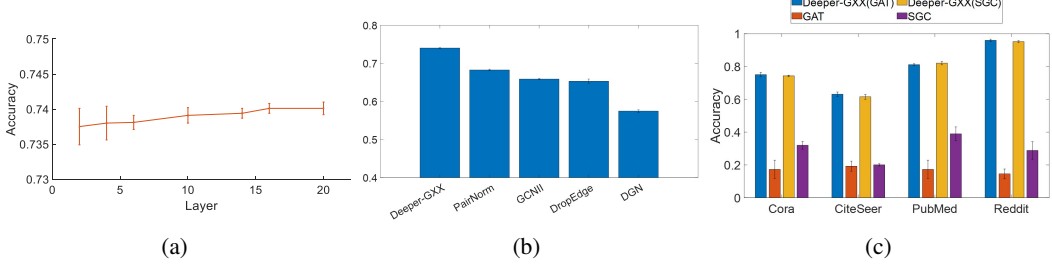

(a)              (b)              (c)

Figure 3: (a) Accuracy of Deeper-GXX on OBG-Arxiv data set with different number of layers. (b) Performance (i.e., accuracy) comparison on OGB-Arxiv data set. (c) Accuracy of different base models with 60 hidden layers on four data sets.

In Figure 3c, we show the performance of our proposed method with different base models (*e.g.*, GAT (Velickovic et al., 2018) and SGC (Wu et al., 2019)). In the experiment, we set the numbers of the hidden layers as 60 for all methods and the dimension of the hidden layer as 50. The total number of training iteration is 1500. By observation, we find that both GAT and SGC suffer from vanishing gradient and over-smoothing when the architecture becomes deeper, and our proposed method Deeper-GXX greatly alleviates them and boosts the performance by 40%-60% on average over four data sets. Specifically, compared with the vanilla SGC, our Deeper-GXX boosts its performance by 43% on the CiteSeer data set and more than 67% on the Reddit data set.

### 3.3   CASE STUDY: MISSING FEATURE SCENARIO

*Why do we need to stack more layers of GNNs?* To answer this question, let us first imagine a scenario where some values of attributes are missing in the input graph. In this scenario, the shallow GNNs may not work well because GNNs could not collect useful information from the neighbors due to the massive missing values. However, if we increase the number of layers, GNNs are able to gather more information from the $k$-hop neighbors and capture latent knowledge to compensate for missing features. To verify this, we conduct the following experiment: we randomly mask $p\%$ attributes in Cora and CiteSeer data sets (i.e., setting the masked attributes to be 0), gradually increase the number of layers, and record the accuracy for each setting. In this case study, the number of layers

Table 2: Accuracy of node classification on two data sets by masking $p$ percent of node attributes. $\#L$ denotes the number of layers where a model achieves the best performance.

| Node Feature Missing Rate | | $p = 25\%$ | | $p = 50\%$ | | $p = 75\%$ | |
|---|---|---|---|---|---|---|---|
| data set | Method | Acc | #L | Acc | #L | Acc | #L |
| Cora | GCN + ResNet | $0.7503 \pm 0.0101$ | 7 | $0.7435 \pm 0.0048$ | 10 | $0.7226 \pm 0.0099$ | 10 |
| | PairNorm + ResNet | $0.7529 \pm 0.0129$ | 10 | $0.7482 \pm 0.0172$ | 20 | $0.7262 \pm 0.0178$ | 40 |
| | DropEdge + ResNet | $0.7634 \pm 0.0112$ | 15 | $0.7611 \pm 0.0102$ | 20 | $0.7297 \pm 0.0168$ | 8 |
| | GCNII + ResNet | $0.2667 \pm 0.0063$ | 25 | $0.3351 \pm 0.0066$ | 25 | $0.2914 \pm 0.0106$ | 40 |
| | DGN w/o ResNet | $0.6850 \pm 0.0184$ | 30 | $0.6846 \pm 0.0147$ | 50 | $0.6717 \pm 0.0156$ | 25 |
| | Deeper-GXX-S | $0.7872 \pm 0.0128$ | 15 | $0.7811 \pm 0.0147$ | 20 | $0.7586 \pm 0.0121$ | 60 |
| | Deeper-GXX | $\mathbf{0.7915 \pm 0.0060}$ | 10 | $\mathbf{0.7848 \pm 0.0043}$ | 20 | $\mathbf{0.7598 \pm 0.0081}$ | 60 |
| CiteSeer | GCN + ResNet | $0.6141 \pm 0.0080$ | 4 | $0.5811 \pm 0.0093$ | 10 | $0.5149 \pm 0.0173$ | 9 |
| | PairNorm + ResNet | $0.6184 \pm 0.0087$ | 8 | $0.5947 \pm 0.0083$ | 20 | $0.5176 \pm 0.0075$ | 10 |
| | DropEdge + ResNet | $0.6348 \pm 0.0156$ | 4 | $0.6083 \pm 0.0128$ | 6 | $0.5240 \pm 0.0128$ | 10 |
| | GCNII + ResNet | $0.2453 \pm 0.0045$ | 40 | $0.2338 \pm 0.0028$ | 20 | $0.2403 \pm 0.0046$ | 25 |
| | DGN w/o ResNet | $0.4560 \pm 0.0162$ | 20 | $0.4593 \pm 0.0117$ | 15 | $0.4498 \pm 0.0292$ | 15 |
| | Deeper-GXX-S | $0.6508 \pm 0.0060$ | 10 | $0.6132 \pm 0.0042$ | 15 | $0.5544 \pm 0.0138$ | 20 |
| | Deeper-GXX | $\mathbf{0.6524 \pm 0.0087}$ | 20 | $\mathbf{0.6169 \pm 0.0063}$ | 60 | $\mathbf{0.5576 \pm 0.0070}$ | 50 |

is selected from $\{2, 3, 4, 5, 6, 7, 8, 9, 10, 15, 20, 25, 30, 40, 50, 60\}$, and the base model is GCN. For fair comparison, we add ResNet (He et al., 2016) if it can boost the baseline model's performance by avoiding the vanishing gradient issue. We repeat the experiments five times and record the mean accuracy and standard deviation.

Table 2 shows the performance of Deeper-GXX and various baselines with the optimal number of layers denoted as #L, i.e., when the model achieves the best performance. By observation, we find that when the missing rate is 25%, shallow GCN with ResNet has enough capacity to achieve the best performance on both CiteSeer and Cora data sets. Compared with GCN, our proposed method further improves the performance by more than 3.83% on the CiterSeer data set and 4.08% on the Cora data set by stacking more layers. However, when we increase the missing rate to 50% and 75%, we observe that most methods tend to achieve the best performance by stacking more layers. Specifically, PairNorm achieves the best performance at 10 layers when 25% features are missing, while it has the best performance at 40 layers when 75% features are missing. A similar observation could also be found with GCNII on the Cora data set, DropEdge on CiteSeer data set as well as our proposed methods in both data sets. Overall, the experimental results verify that the more features a data set is missing, the more layers GNNs need to be stacked to achieve better performance. Our explanation for this observation is that if the number of layers increases, more information will be collected from the $k$-hop neighbors to recover the missing information of its 1-hop and 2-hop neighbors.

## 3.4 ABLATION STUDY

In this subsection, we conduct an ablation study on Cora to show the effectiveness of WDG-ResNet and TGCL in Table 3. In this experiment, we fix the feature dimension of the hidden layer as 50, the total iteration is set as 3000, the number of layers is set as 60, the sampling batch size for Deeper-GXX is 10, and GCN is chosen as the base model. In Table 3, Deeper-GXX-T removes TGCL loss, Deeper-GXX-D removes the weight decaying factor in WDG-ResNet and

Table 3: Ablation Study on Cora Data Set w.r.t Node Classification Accuracy

| Method | Accuracy |
|---|---|
| Deeper-GXX | $\mathbf{0.8059 \pm 0.0028}$ |
| Deeper-GXX-S | $0.8023 \pm 0.0117$ |
| Deeper-GXX-D | $0.7498 \pm 0.0139$ |
| Deeper-GXX-T | $0.7875 \pm 0.0092$ |

Deeper-GXX-S achieves the simplified WDG-ResNet in Deeper-GXX, which removes the similarity measure in WDG-ResNet. In Table 3, we have the following observations (1) comparing Deeper-GXX with Deeper-GXX-T, we find that Deeper-GXX boosts the performance by 1.84% after adding TGCL loss, which demonstrates the effectiveness of TGCL to address over-smoothing issue; (2) Deeper-GXX outperforms Deeper-GXX-D by 5.61%, which shows that Deeper-GXX could address the shading neighbors effect by adding the weight decaying factor; (3) comparing Deeper-GXX with Deeper-GXX-S, we verify that adding exponential cosine similarity measure $e^{cos(\boldsymbol{H}^{(1)}, \tilde{\boldsymbol{H}}^{(l)})}$ could further improve the performance by extending the hypothesis space of neural networks.

## 4 RELATED WORK

**Contrastive Learning on Graphs**. Recently, contrastive learning attracts researchers' great attention due to its outstanding performance by leveraging the rich unsupervised data. (van den Oord et al., 2018) is one of the earliest works, which proposes a Contrastive Predictive Coding framework to extract useful information from high dimensional data with a theoretical guarantee. Based on this work, recent studies (Song & Ermon, 2020; Chuang et al., 2020; Khosla et al., 2020; Tian et al., 2020; Chen et al., 2020c; Zheng et al., 2022) reveal a surge of research interest in contrastive learning. (You et al., 2020) proposes a graph contrastive learning (GraphCL) framework that utilizes different types of augmentation methods to incorporate various priors and to learn unsupervised representations of graph data. (Qiu et al., 2020) proposes a Graph Contrastive pre-training model named GCC to capture the graph topological properties across multiple networks by utilizing contrastive learning to learn the intrinsic and transferable structural representations. (Zheng et al., 2021) introduced a weakly supervised contrastive learning framework to tackle the class collision problem by first generating a weak label for similar samples and then pulling the similar samples closer with contrastive regularization. Authors Hassani & Ahmadi (2020) aims to learn node and graph level representations by contrasting structural views of graphs. In this paper, we leverage the contrastive learning manner and design positive and negative node pairs such that we can discriminate node representations based on the input graph inherent information, which paves the way for us to design effective deeper GNNs.

**Deeper Graph Neural Networks**. To effectively make GNNs deeper, many research works focus on addressing the over-smoothing problem. Over-smoothing problem of GNNs is formally described by (Li et al., 2018) to demonstrate that the final output node representations become indiscriminative after stacking many layers in GNN models. This problem is also analyzed by (Oono & Suzuki, 2020) showing how over-smoothing hurts the node classification performance. To quantify the degree of over-smoothing, different measurements are proposed (Chen et al., 2020a; Zhao & Akoglu, 2020; Liu et al., 2020; Zhou et al., 2020). For example, Mean Average Distance (Chen et al., 2020a) is proposed by calculating the divergences between learned node representations. To make GNNs deeper and maintain performance, some nascent research works are proposed (Klicpera et al., 2019; Chen et al., 2020a; Zhao & Akoglu, 2020; Rong et al., 2020; Chen et al., 2020b; Liu et al., 2020; Zhou et al., 2020). Among those, some of them share the same logic of keeping the divergence between node representations but differ in specific methodologies. For example, PairNorm (Zhao & Akoglu, 2020) introduces a normalization layer to keep the divergence of node representation from the original input node feature. In DGN (Zhou et al., 2020), node representation by deep GNNs is regularized by group-based mutual information. However, the hyperparameters selection is important and data-driven, which means many efforts need to be paid for the hyperparameter searching. Some methods focus on changing the information aggregation scheme to make GNNs deeper, such as APPNP (Klicpera et al., 2019), GCNII (Chen et al., 2020b), DropEdge (Rong et al., 2020) and etc. To the best of our knowledge, in dealing with over-smoothing problem, our Deeper-GXX is the first attempt to transfer hard-to-acquire discriminative knowledge into the topology information of the input graph by comparing adjacency vectors of nodes. Also, we analyze another problem (i.e., vanishing gradient) that hinders GNNs to be effectively deeper and discover the *shading neighbors* effect caused by simply applying ResNet on GNNs, and propose a GNN-based residual connection to avoid this issue and improve the performance of deeper GNNs.

## 5 CONCLUSION

In this paper, we focus on building deeper graph neural networks to effectively model graph data. To this end, we first provide insights regarding why ResNet is not best suited for many deeper GNN solutions, i.e., the shading neighbors effect. Then we propose a new residual architecture, Weight-Decaying Graph Residual Connection (WDG-ResNet) to address this effect. In addition, we propose a Topology-guided Graph Contrastive Loss (TGCL) to address the problem of over-smoothing, where we utilize graph topological information, pull the representations of connected node pairs closer, and push remote node pairs farther apart via contrastive learning regularization. Combining WDG-ResNet with TGCL, an end-to-end model named Deeper-GXX is proposed towards deeper GNNs. We provide the theoretical analysis of our proposed method, and demonstrate the effectiveness of Deeper-GXX by extensive experiment comparing with state-of-the-art de-oversmoothing algorithms. A case study regarding the missing feature scenario demonstrates the necessity to deepen the GNNs.

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

## A  EXPERIMENT

### A.1  CASE STUDY: A TOY EXAMPLE

*Why do we need to stack more layers of GNNs?* To answer this question, we conduct the experiment in a toy example. We first use the existing package (i.e., draw circle function in the Scikit-learn package) to generate a synthetic data set by setting the number of data points to be 1,000 and the noise level to be 0.01. Then, we measure the euclidean distance between each pair of data points, and if the the distance is less than a threshold, then this two data points are connected in a graph. In this way, the adjacency matrix is derived after adding the self loop. Next, we sample 1% data points as the training set, 9% data points as the validation set, and 90% data points as the test set. These data points are visualized in Figure 1a and the experimental results are shown in Figure 1b. In Figure 1a, we observe that the query node (the diamond in the red dashed circle) cannot rely on its closest labeled neighbor (the red star in the circle) to correctly predict its label (red or blue). Only by exploring longer paths consisting of more similar neighbors are we able to predict its label as blue. Figure 1b compares the classification accuracy of shallow GNNs and deeper GNNs. We can see that deeper GNNs significantly outperform shallow ones by more than 11%, due to their abilities to explore longer paths on the graph.

### A.2  EXPERIMENTAL SETTING

In this subsection, we provide the detailed experimental setting for each experiment shown in Table 4.

Table 4: Hyperparameters for Deeper-GXX and Deeper-GXX-S shown in Table 1

| Method | Deeper-GXX | Deeper-GXX-S |
|--------|------------|--------------|
| Cora | $\lambda = 20, \alpha = 0.03$ | $\alpha = 0.01$ |
| CiteSeer | $\lambda = 10, \alpha = 0.02$ | $\alpha = 0.01$ |
| PubMed | $\lambda = 18, \alpha = 0.1$ | $\alpha = 0.1$ |
| Reddit | $\lambda = 20, \alpha = 0.02$ | $\alpha = 0.01$ |

### A.3  MORE RESULTS IN ABLATION STUDY

In addition to the ablation study in Section 3.4, here in Table 5, we conduct more ablation experiments on CiteSeer, PubMed, Reddit, and OGB-Arxiv data sets, following the same experimental setting in Table 3, to demonstrate the effectiveness of each component in Deeper-GXX by comparing it with three variants.

Table 5: Ablation Study on CiteSeer, PubMed, Reddit, and OGB-Arxiv Data Sets w.r.t. Node Classification Accuracy

| Method | CiteSeer | PubMed | Reddit | OGB-Arxiv |
|--------|----------|--------|--------|-----------|
| Deeper-GXX | **0.6655 ± 0.0117** | 0.8185 ± 0.0016 | **0.9721 ± 0.0011** | **0.7401 ± 0.0009** |
| Deeper-GXX-S | 0.6544 ± 0.0099 | **0.8198 ± 0.0012** | 0.9693 ± 0.0036 | 0.7382 ± 0.0008 |
| Deeper-GXX-D | 0.6567 ± 0.0052 | 0.8150 ± 0.0031 | 0.9654 ± 0.0028 | 0.7363 ± 0.0011 |
| Deeper-GXX-T | 0.5750 ± 0.0244 | 0.8098 ± 0.0047 | 0.9397 ± 0.0042 | 0.7335 ± 0.0024 |

### A.4  VISUALIZATION OF WEIGHT OF EACH LAYER WITH DIFFERENT WEIGHTING FUNCTIONS

In this subsection, we visualize the weight of each layer with different weighting functions on the Cora data set. In this experiment, we fix the feature dimension of the hidden layer to be 50; the total iteration is set to be 3000; the number of layers is set to be 60; the sampling batch size for Deeper-GXX is 10; GCN is chosen as the base model; $\lambda$ is set to be 20. In Figure 4, The x-axis is the index of each layer and the y-axis is the weight for each layer. Deeper-GXX-D removes the decaying weight factor and only keeps the exponential cosine similarity $e^{cos(\boldsymbol{H}^{(1)}, \tilde{\boldsymbol{H}}^{(l)})}$ to measure the weight for each layer. Deeper-GXX-S achieves the simplified WDG-ResNet in Deeper-GXX,

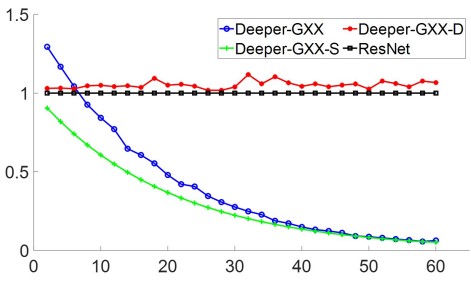

Figure 4: Weight visualization. The y-axis represents the weight of each layer, and x-axis represents the index of each layer, in deeper models

which removes the exponential cosine similarity $e^{cos(\boldsymbol{H}^{(1)},\tilde{\boldsymbol{H}}^{(l)})}$ in Deeper-GXX. By observation, we find that (1) ResNet sets the weight of each layer to be 1, which easily leads to *shading neighbors* effect when stacking more layers, because the faraway neighbor information becomes more dominant in the GCN information aggregation; (2) without weight decaying factor, the weight for each layer in Deeper-GXX-D fluctuates because they are randomly independent. More specially, the weights for the last several layers (e.g., L=58 or L=60) are larger than the weights for the first several layers, which contradicts the intuition that the first several layers should be important than the last several layers; (3) the weights for each layer in both Deeper-GXX and Deeper-GXX-S reduce as the number of layers increase, which suggests that both of them could address the *shading neighbors* effect to some extents; (4) combining the results from Table 3, Deeper-GXX achieves better performance than Deeper-GXX-S, as it imposes larger weights to the first several layers, which verifies that the learnable similarity $sim(\boldsymbol{H}^{(1)},\tilde{\boldsymbol{H}}^{(l)})$ achieves better performance with the enlarged hypothesis space for neural networks.

## A.5   HYPERARAMETER ANALYSIS

In this subsection, we conduct the hyperparameter analysis of Deeper-GXX, regarding $\lambda$ in the weight decaying function of Eq. 4 and $\alpha$ in the overall loss function of Eq 1.

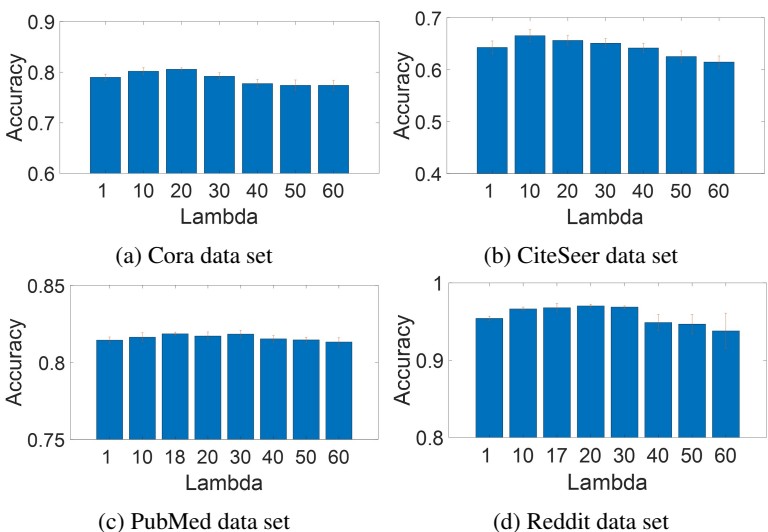

(a) Cora data set      (b) CiteSeer data set

(c) PubMed data set      (d) Reddit data set

Figure 5: Hyperparameter analysis, i.e., $\lambda$ vs accuracy score on four data sets

To analyze the hyperparameter $\lambda$, we fix the feature dimension of the hidden layer to be 50, the total iteration is set to be 3000, the number of layers is set to be 60, the sampling batch size for Deeper-GXX is 10, and GCN is chosen as the base model. The experiment is repeated five times for each configuration. In each sub-figure of Figure 5, the x-axis is the value of $\lambda$, and the y-axis

is the accuracy of 60-layer GCN in the above setting. First, we can observe that it's not true that Deeper-GXX achieves the best performance with a larger $\lambda$. Specifically, we find that the optimal $\lambda = 20$ on Cora data set, the optimal $\lambda = 10$ on CiteSeer data set, the optimal $\lambda = 18$ on PubMed data set, and the optimal $\lambda = 20$ on Reddit data set. Then, natural questions to ask are *(1) what determines the optimal value of $\lambda$ in different data sets? (2) can we gather some heuristics to narrow down the hyperparameter search space to efficiently establish effective GNNs*? Here, we provide our discovery. In the main text of the paper, we have analyzed that the decaying factor $\lambda$ in Eq. 4 controls the number of effective layers in deeper GNNs by introducing the layer-wise dependency. It means that larger $\lambda$ slows down the weight decay and gives considerable large weights to more layers such that they can be effective, and the information aggregation scope of GNN extends as more multi-hop neighbors features are collected and aggregated. In graph theory, diameter represents the scope of the graph, which is the largest value of the shortest path between any node pairs in the graph. Therefore, the optimal $\lambda$ should be restricted by the input graph, i.e., being close to the input graph diameter. Interestingly, our experiments reflect this observation. Combining the optimal $\lambda$ in Figure 5 and the diameter in Table 6, for connected graphs PubMed and Reddit, the optimal $\lambda$ is very close to the graph diameter. This also happens to Cora (even though Cora is not connected), because the number of components is not large. As for CiteSeer, the optimal $\lambda$ is less than the diameter of its largest component. A possible reason is that CiteSeer has many (i.e., 438) small components, which shrinks the information propagation scope, such that we do not need to stack many layers and we do not need to enlarge $\lambda$ to the largest diameter (i.e., 28). In general, based on the above analysis, we find the optimal value of $\lambda$ can be searched around the diameter of the input graph.

Table 6: Graph Statistics of each Data Set

|  | Cora | Citeseer | PubMed | Reddit |
|---|---|---|---|---|
| Number of nodes | 2,708 | 3,327 | 19,717 | 4,854 |
| Connected graph or not | No | No | Yes | Yes |
| Number of components | 78 | 438 | 1 | 1 |
| Diameter of the graph or the largest component | 19 | 28 | 18 | 17 |

To analyze the hyperparameter $\alpha$ in Deeper-GXX, we fix the feature dimension of the hidden layer to be 50, the total iteration is set to be 3000, the number of layers is set to be 60, the sampling batch size for Deeper-GXX is 10, GCN is chosen as the base model, and the data set is Cora. We gradually increase the value of $\alpha$ and record the accuracy. The experiment is repeated five times in each setting. In Figure 6, the x-axis is $\alpha$ and the y-axis is the accuracy score. By observation, when $\alpha = 1$, the performance is worst and the performance begins to increase by decreasing the value of $\alpha$. It achieves the best accuracy when $\alpha = 0.03$. The performance starts to decrease again if we further decrease the value of $\alpha$. Our conjecture is that when $\alpha$ is

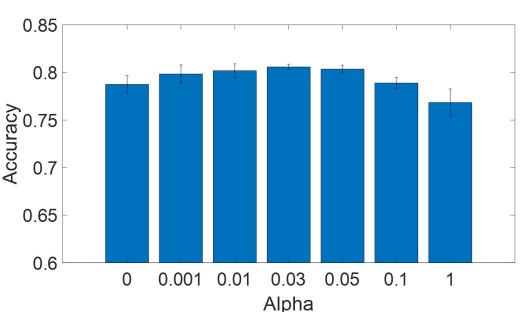

Figure 6: Hyperparameter analysis, i.e., $\alpha$ vs accuracy score

large, it will dominate the overall objective function, thus jeopardizing the classification performance. Besides, if we set the value of $\alpha$ to be a small number (i.e., $\alpha = 0.001$), the performance also decreases. In addition, comparing with the performance without using TGCL regularization (i.e., $\alpha = 0$), our proposed method with $\alpha = 0.03$ can boost the performance by more than 1.8%, which demonstrates that our proposed TGCL alleviates the issue of oversmoothing to some extent.

## A.6 EFFICIENCY ANALYSIS

In this subsection, we conduct an efficiency analysis regarding our proposed method in the Cora data set. We fix the feature dimension of the hidden layer to be 50, the total iteration is set to be 1500, the sampling batch size for Deeper-GXX and Deeper-GXX-S is 10, and GCN is chosen as the base model. We gradually increase the number of layers and record the running time. In Figure 7,

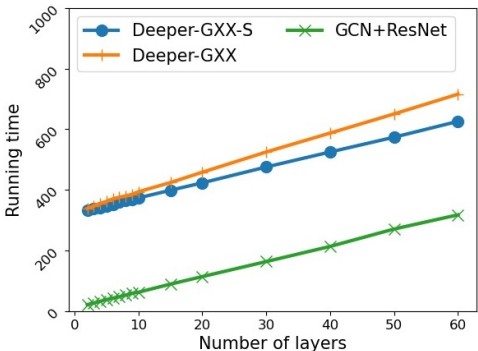

Figure 7: The number of layers vs running time (in seconds) on Cora data set. (Best viewed in color)

the x-axis is the number of layers and the y-axis is the running time in second. We observe that the running time of both Deeper-GXX and Deeper-GXX-S is linearly proportional to the number of layers. Comparing the running time of Deeper-GXX, the running time of Deeper-GXX-S is further reduced after the weighting function in Deeper-GXX (e.g., $sim(\cdot)$) is replaced by a constant.

## B  ANALYSIS

### B.1  PROOF OF PROPOSITION 1

**Proof 1** *Following the theoretical analysis in  (van den Oord et al., 2018) Section 2.3, the optimal value of $f(\boldsymbol{z}_i, \boldsymbol{z}_j)$ is given by $\frac{P(\boldsymbol{z}_j|\boldsymbol{z}_i)}{P(\boldsymbol{z}_j)}$. Thus, the weighted supervised contrastive loss could be rewritten as follows:*

$$\mathcal{L}_{\text{TGCL}} = - \mathbb{E}_{v_i \sim V} \mathbb{E}_{v_j \in \mathcal{N}_i} [\log \frac{\sigma_{ij} f(\boldsymbol{z}_i, \boldsymbol{z}_j)}{\sigma_{ij} f(\boldsymbol{z}_i, \boldsymbol{z}_j) + \sum_{v_k \in \bar{\mathcal{N}}_i} \gamma_{ik} f(\boldsymbol{z}_i, \boldsymbol{z}_k)}]$$

$$= \mathbb{E}_{v_i \sim V} \mathbb{E}_{v_j \in \mathcal{N}_i} [\log \frac{\sigma_{ij} \frac{P(\boldsymbol{z}_j|\boldsymbol{z}_i)}{P(\boldsymbol{z}_j)} + \sum_{v_k \in \bar{\mathcal{N}}_i} \gamma_{ik} \frac{P(\boldsymbol{z}_k|\boldsymbol{z}_i)}{P(\boldsymbol{z}_k)}}{\sigma_{ij} \frac{P(\boldsymbol{z}_j|\boldsymbol{z}_i)}{P(\boldsymbol{z}_j)}}]$$

$$= \mathbb{E}_{v_i \sim V} \mathbb{E}_{v_j \in \mathcal{N}_i} [\log(1 + \frac{P(\boldsymbol{z}_j)}{\sigma_{ij} P(\boldsymbol{z}_j|\boldsymbol{z}_i)} \sum_{v_k \in \bar{\mathcal{N}}_i} \gamma_{ik} \frac{P(\boldsymbol{z}_k|\boldsymbol{z}_i)}{P(\boldsymbol{z}_k)})]$$

*Since $(v_i, v_k)$ is defined as a remote (i.e., negative) node pair, it means that node $v_i$ and node $v_k$ are not connected in the graph, i.e., $A_{i,k} = A_{k,i} = 0$. Therefore, we have $\gamma_{ik} \in (1, 2]$ for all remote nodes $v_k$ and $\sigma_{ij} \in (0, 1]$ for all neighbor nodes $v_j$ with hamming distance measurement, which leads to $\frac{1}{\sigma_{ij}} \cdot \frac{P(\boldsymbol{z}_j)}{P(\boldsymbol{z}_j|\boldsymbol{z}_i)} \geq \frac{P(\boldsymbol{z}_j)}{P(\boldsymbol{z}_j|\boldsymbol{z}_i)}$ and $\gamma_{ik} \frac{P(\boldsymbol{z}_k|\boldsymbol{z}_i)}{P(\boldsymbol{z}_k)} \geq \frac{P(\boldsymbol{z}_k|\boldsymbol{z}_i)}{P(\boldsymbol{z}_k)}$. Thus, we have*

$$\mathcal{L}_{\text{TGCL}} \geq \mathbb{E}_{v_i \sim V} \mathbb{E}_{v_j \in \mathcal{N}_i} [\log(\frac{P(\boldsymbol{z}_j)}{P(\boldsymbol{z}_j|\boldsymbol{z}_i)} \sum_{v_k \in \bar{\mathcal{N}}_i} \frac{P(\boldsymbol{z}_k|\boldsymbol{z}_i)}{P(\boldsymbol{z}_k)})]$$

$$\approx \mathbb{E}_{v_i \sim V} \mathbb{E}_{v_j \in \mathcal{N}_i} [\log(\frac{P(\boldsymbol{z}_j)}{P(\boldsymbol{z}_j|\boldsymbol{z}_i)} (|\bar{\mathcal{N}}_i| \mathbb{E}_{v_k} \frac{P(\boldsymbol{z}_k|\boldsymbol{z}_i)}{P(\boldsymbol{z}_k)}))]$$

$$= \mathbb{E}_{v_i \sim V} \mathbb{E}_{v_j \in \mathcal{N}_i} [\log(\frac{P(\boldsymbol{z}_j)}{P(\boldsymbol{z}_j|\boldsymbol{z}_i)} |\bar{\mathcal{N}}_i|)]$$

$$\geq \mathbb{E}_{v_i \sim V} \mathbb{E}_{v_j \in \mathcal{N}_i} [\log(\frac{P(\boldsymbol{z}_j)}{P(\boldsymbol{z}_j|\boldsymbol{z}_i)}) + \log(|\bar{\mathcal{N}}_i|)]$$

$$= -I(\boldsymbol{z}_i, \boldsymbol{z}_j) + \mathbb{E}_{v_i \sim V} \log(|\bar{\mathcal{N}}_i|)$$

*Finally, we have $I(\boldsymbol{z}_i, \boldsymbol{z}_j) \geq -\mathcal{L}_{\text{TGCL}} + \mathbb{E}_{v_i \sim V} \log(|\bar{\mathcal{N}}_i|)$, which completes the proof.*

## B.2    SAMPLING METHOD FOR TGCL

To realize TGCL loss function expressed in Eq. 6, we need to get the positive nodes $v_j$ and negative nodes $v_k$ towards the selected central node $v_i$. To avoid iterating over all existing nodes or randomly sampling several nodes, we propose to sample positive nodes $v_j$ and negative nodes $v_k$ from the star subgraph $S_i$ of the central node $v_i$. Moreover, to make the sampling be scalable and to reduce the search space of negative nodes, we propose a batch sampling method.

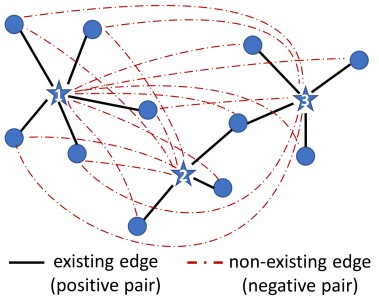

existing edge
(positive pair)

non-existing edge
(negative pair)

As shown in Figure 8, the batch size is controlled by the number of central nodes (i.e., star nodes in the figure). For each central node, the positive nodes are those 1-hop neighbors, and the negative nodes consist of unreachable nodes. In our batch sampling, we strictly constrain that the positive nodes are only from the 1-hop neighborhood for the following three reasons: (1) they are efficient to be accessed; (2) considering all $k$-hop neighbors as positive will enlarge the scope of positive nodes and further decrease the intimacy of the directly connected nodes; (3) 1-hop positive nodes in the star subgraph can preserve enough useful information, compared with the positive nodes from the whole graph. For the third point, we prove it through the *graph influence loss* (Huang & Zitnik, 2020) in Proposition 2, and the formal definition of *graph influence loss* is given in the following paragraph after Proposition 2.

Figure 8: Batch Sampling. Each star node in the figure corresponds to node $v_i$ in Eq. 6.

**Proposition 2 (Bounded Graph Influence Loss for Sampling Positive Pairs Locally)** *Taking GCN as an example of GNN, the graph influence loss $R(v_c)$ on node $v_c$ w.r.t **positive nodes from the whole graph** against **positive nodes from the 1-hop neighborhood star subgraph** is bounded by $R(v_c) \leq (n - d_c)\frac{\mu}{(D_{GM}^{\bar{\mathcal{P}}_*})^{|\bar{\mathcal{P}}_*|}}$ , where $n$ is the number of nodes, $d_c$ is the degree of node $v_c$ including the self-loop, $\mu$ is a constant, $\bar{\mathcal{P}}_*$ is the path from center node $v_c$ to a 1-hop outside node $v_s$ which has the maximal node influence $I_{v_c,v_s}$, and $|\bar{\mathcal{P}}_*|$ denotes the number of nodes in path $\bar{\mathcal{P}}_*$.*

**Proof 2** *According to the assumption of (Wang & Leskovec, 2020), $\sigma(\cdot)$ can be identity function and $\boldsymbol{W}^{(\cdot)}$ can be identity matrix. Then, the hidden node representation (of node $v_c$) in the last layer of GCN can be written as follows.*

$$\boldsymbol{h}_c^{(\infty)} = \frac{1}{D_{c,c}} \sum_{v_i \in \mathcal{N}_c} A_{c,i} \boldsymbol{h}_i^{(\infty)}$$

*Then, based on the above equation, we can iteratively replace $\boldsymbol{h}_i^{(\infty)}$ with its neighbors until the representation $\boldsymbol{h}_s^{(\infty)}$ of node $v_s$ is included. The extension procedure is written as follows.*

$$\boldsymbol{h}_c^{(\infty)} = \frac{1}{D_{c,c}} \sum_{v_i \in \mathcal{N}_c} A_{c,i} \frac{1}{D_{i,i}} \sum_{v_j \in \mathcal{N}_i} A_{i,j} \cdots \frac{1}{D_{k,k}} \sum_{v_s \in \mathcal{N}_k} A_{k,s} \boldsymbol{h}_s^{(\infty)}$$

*The above equation suggests that the influence from the positive node $v_s$ to the center node $v_c$ is through the path $\mathcal{P} = (v_c, v_i, v_j, \ldots, v_k, v_s)$.*

*Following the above path formation and assume the edge weight $A(i, j)$ as the positive constant, according to (Huang & Zitnik, 2020), we can obtain the node influence $I_{v_c,v_s}$ of $v_s$ on $v_c$ as follows.*

$$I_{v_c,v_s} = \|\partial \boldsymbol{h}_c^{(\infty)} / \partial \boldsymbol{h}_s^{(\infty)}\| \leq \frac{\mu}{(D_{GM}^{\bar{\mathcal{P}}})^{|\bar{\mathcal{P}}|}}$$

*where $\mu$ is a constant, $D_{GM}^{\bar{\mathcal{P}}}$ is the geometric mean of degree of nodes sitting in path $\bar{\mathcal{P}}$, and $\bar{\mathcal{P}}$ is the path from the positive node $v_s$ to the center node $v_c$ that could generate the maximal multiplication of normalized edge weight, $|\bar{\mathcal{P}}|$ denotes the number of nodes in path $\bar{\mathcal{P}}$.*

*The above analysis suggests that the node influence of long-distance positive nodes is decaying.*

*Hence, the graph influence loss about learning node $v_c$ from **the whole graph positive nodes** versus from the **1-hop localized positive nodes** can be expressed expressed as follows.*

$$
\begin{aligned}
I_G(v_c) - I_L(v_c) = I_{v_c,v_1} + I_{v_c,v_2} + \ldots + I_{v_c,v_{n-d_c}} &\leq \sum_{i=1}^{n-d_c} \frac{\mu_i}{(D_{GM}^{\bar{\mathcal{P}}_i})^{|\bar{\mathcal{P}}_i|}} \\
&\leq (n - d_c) \frac{\mu_*}{(D_{GM}^{\bar{\mathcal{P}}_*})^{|\bar{\mathcal{P}}_*|}}
\end{aligned}
$$

*where $I_G(v_c)$ denotes global influence, $I_L(v_c)$ is the influence for star subgraph, $d_c$ is the degree of node $v_c$ (including self-loop), and $\frac{\mu_*}{(D_{GM}^{\bar{\mathcal{P}}_*})^{|\bar{\mathcal{P}}_*|}}$ is the maximal among all $\frac{\mu_i}{(D_{GM}^{\bar{\mathcal{P}}_i})^{|\bar{\mathcal{P}}_i|}}$.*

Specifically, the graph influence loss (Huang & Zitnik, 2020) $R(v_c)$ can be expressed as $R(v_c) = I_G(v_c) - I_L(v_c)$, which is determined by the global graph influence on $v_c$ (i.e., $I_G(v_c)$) and the star subgraph influence on $v_c$ (i.e., $I_L(v_c)$). Then, to compute the graph influence $I_G(v_c)$, we need to compute node influence of each node $v_j$ to node $v_c$, where node $v_j$ is reachable from node $v_c$. Based on the final output node representation vectors, the node influence is expressed as $I_{v_c,v_j} = \|\partial \boldsymbol{h}_c^{(\infty)} / \partial \boldsymbol{h}_j^{(\infty)}\|$, and the norm can be any subordinate norm (Wang & Leskovec, 2020). Then, $I_G(v_c)$ is computed by the $L1$-norm of the following vector, i.e., $I_G(v_c) = \|[I_{v_c,v_1}, I_{v_c,v_2}, \ldots, I_{v_c,v_n}]\|_1$. Similarly, we can compute the star subgraph influence $I_L(v_c)$ on node $v_c$. The only difference is that we collect each reachable node $v_j$ in the star subgraph $L$ (i.e., 1-hop neighbours of $v_c$). Overall, in Proposition 2, we show why positive pairs can be locally sampled with the support from graph influence loss of a node representation vector output by the GCN final layer.

