# OpenReview forum: "DEEPER-GXX: DEEPENING ARBITRARY GNNS"
_ICLR.cc/2023/Conference — Submitted to ICLR 2023_

### Official Review · Reviewer_G8Tr · 2022-10-25

**Confidence:** 4
**Correctness:** 3
**Technical Novelty And Significance:** 2
**Empirical Novelty And Significance:** 2
**Recommendation:** 5

**Clarity, Quality, Novelty And Reproducibility:**

The work is easy to follow. No code was provided but the experimental results should not be difficult to reproduce. The technical novelty of the paper is marginal compared to previous work on deepening GNNs.

**Strength And Weaknesses:**

Pros:

The proposed Weight-Decaying Graph Residual Connection is interesting.

Cons:

Why is the graph residual connection expressed as $H^{(l)} = \sigma(\hat{A} H^{(l-1)} W^{(l-1)}) + H^{(l-2)}$ in Equation 2? Is it typically $H^{(l)} = \sigma(\hat{A} H^{(l-1)} W^{(l-1)}) + H^{(l-1)}$?

The proposed Topology-Guided Graph Contrastive Loss which considers 1-hop neighbors as the positive pair and the rest as negative pairs has a strong assumption on homophily. I wonder if the proposed method would be helpful for graphs with strong heterophily.

In my opinion, this work is not that related to contrastive learning on graphs. However, a large paragraph is used to discuss contrastive learning on graphs which is not necessary. I think the related work can be improved to focus more related works on over-smoothing and different techniques in deepening GNNs.

The experiments are done on relatively small datasets which makes the results less conclusive. I wonder if the authors could perform experiments on recent GNN benchmarks such as Open Graph Benchmark and Benchmarking Graph Neural Networks.

=== post rebuttal ====

The authors address some of my concerns. I increased my score. But I still have doubts about the claim that the proposed method can work well on graphs with strong heterophily.

**Summary Of The Paper:**

This paper proposes Weight-Decaying Graph Residual Connection and Topology-Guided Graph Contrastive Loss to overcome the over-smoothing issue of GNNs in order to train deeper GNNs. Experiments are conducted on citation networks and social networks such as Cora CiteSeer PubMed Reddit.

**Summary Of The Review:**

The major issue of acceptance is the evaluation of the proposed method is done on relatively small datasets. It is not clear how significant it is compared to other SOTA deep GNN methods. I recommend the authors should evaluate their methods on Open Graph Benchmark and Benchmarking Graph Neural Networks to make their claims more conniving.

---

> ### Author Response · Authors · 2022-11-14
> **From authors: reply to concerns of reviewer G8Tr (1/2)**
>
> Thanks for your review and interest in our weight-decaying residual connection mechanism. We would like to address your raised concerns as follows. Each concern raised by the reviewer is indexed by numbers, followed by our response.
>
> 1. **Why is the graph residual connection expressed as $H^{(l)}=\sigma(\hat{A}H^{(l−1)}W^{(l−1)})+H^{(l−2)}$ in Equation 2? Is it typically $H^{(l)}=\sigma(\hat{A}H^{(l−1)}W^{(l−1)})+H^{(l−1)}$**?
>
> Typically, the residual connection connects the current layer $l$ with its second last layer $l-2$, according to [He et al., CVPR 2016] and [Wiki page](https://en.wikipedia.org/wiki/Residual_neural_network).
>
>
> 2. **I wonder if the proposed method would be helpful for graphs with strong heterophily**.
>
> We would like to refer the reviewer to the case study in Figure 1, which is a heterophyllous toy graph that motivates our Deeper-GXX. In Figure 1a, for the query node (blue diamond), the close neighbor is marked with a different label (red star), but the faraway neighbor is marked with the same label (blue star). The corresponding performance of our Deeper-GXX with shallow GNNs in this heterophily setting is shown in Figure 1b, which is better than the baselines.
>
> Moreover, we did use large heterophyllous datasets in the paper. Please refer to **the first bullet point** of the top section titled “From authors: reply to shared concerns from different reviewers”. Briefly, we utilized the large dataset OGB-Arxiv with 169,343 nodes and 1,166,243 edges. It is a heterophyllous dataset, whose edge homophily is only 0.222 measured by [Lim et al., GLB@WWW 2021]. And our Deeper-GXX also outperforms these baselines. The details can be found in Subsection 3.1 Experiment Setup - Datasets, and performance, Figures 3a and 3b.
>
> 3. **In my opinion, this work is not related to contrastive learning on graphs. However, a large paragraph is used to discuss contrastive learning on graphs which is not necessary. I think the related work can be improved to focus more related works on over-smoothing and different techniques in deepening GNNs**.
>
> In addition to dealing with vanishing gradient and shading neighbor effects by residual neural connections, another main contribution of Deeper-GXX is to alleviate the over-smoothing problem, which recently attracted much research attention and stays in the nascent stage. The key idea of contrasting is effective for addressing the over-smoothing problem in deeper GNNs. Many works like PairNorm [Zhao et al., ICLR 2020] and DGN [Zhou et al., NeurIPS 2020] depend on contrasting and then develop respective improvements for effectiveness and efficiency.  We will consider your suggestion by reorganizing the flow to emphasize the relationship between contrastive learning and over-smoothing in the camera ready. The detailed reasons are listed below.
>
> PairNorm [Zhao et al., ICLR 2020] introduces the hyperparameter to scale the distance between pair nodes to contrast their hidden representations (i.e., Eq. 11 in PairNorm paper) and DGN [Zhou et al., NeurIPS 2020] introduces the mutual information constraint to a grouping (i.e., similarity measurement) of nodes (i.e., Eq. 5 and 6 in DGN paper). (Note that these two are also baselines in our paper.) For a similar rationale, we develop the simple but effective TGCL contrastive loss in Deeper-GXX, which avoids the hard-to-acquire prior contrastive knowledge like hyperparameters or grouping methods, but transfers them into the easy-to-obtain information, i.e., adjacency vector associated with the proper distance function. The theoretical effectiveness analysis is provided in Appendix B, and the empirical performance is shown in the ablation study in Table 3, where Deeper-GXX-T stands for the removal of TGCL from Deeper-GXX. Also, we would like to invite you to the extended ablation study in Table 5 in Appendix.

---

> > ### Author Response · Authors · 2022-11-14
> > **From authors: reply to concerns of reviewer G8Tr (2/2)**
> >
> > 4. **The experiments are done on relatively small datasets which makes the results less conclusive. I wonder if the authors could perform experiments on recent GNN benchmarks such as Open Graph Benchmarks and Benchmarking Graph Neural Networks**.
> >
> > We would like to point out that we did use the OGB dataset, where we compared Deeper-GXX with SOTA baselines to show its superior performance. Please refer to **the first bullet point** of the top section titled “From authors: reply to shared concerns from different reviewers”.
> >
> > *Not listed as a weakness but the reviewer mentioned no code is provided. We would like to point out that the code was provided in the paper via an anonymous link.*
> >
> > Reference
> >
> > [He et al., CVPR 2016] Kaiming He, Xiangyu Zhang, Shaoqing Ren, Jian Sun: Deep Residual Learning for Image Recognition. CVPR 2016
> >
> > [Lim et al., GLB@WWW 2021] Derek Lim, Xiuyu Li, Felix Hohne, Ser-Nam Lim: New Benchmarks for Learning on Non-Homophilous Graphs. GLB@WWW 2021
> >
> > [Zhao et al., ICLR 2020] Lingxiao Zhao, Leman Akoglu: PairNorm: Tackling Oversmoothing in GNNs. ICLR 2020
> >
> > [Zhou et al., NeurIPS 2020] Kaixiong Zhou, Xiao Huang, Yuening Li, Daochen Zha, Rui Chen, Xia Hu: Towards Deeper Graph Neural Networks with Differentiable Group Normalization. NeurIPS 2020

---

> ### Author Response · Authors · 2022-11-18
> **From authors: discussions are welcome**
>
> Dear Reviewer,
>
> Until now we have not received your discussion and the time will be up soon, would you please discuss based on what we replied?
>
> Best,
>
> Authors

---

> > ### Author Response · Authors · 2022-12-09
> > **From authors: discussions are still welcome**
> >
> > Dear Reviewer,
> >
> > The second discussion phase seems to come to an end very soon, we still did not know how you evaluate our response answer for your raised concerns. We sincerely thank you for your time and hard work in advance!
> >
> > Best,
> >
> > Authors

---

### Official Review · Reviewer_9Psf · 2022-10-25

**Confidence:** 4
**Correctness:** 2
**Technical Novelty And Significance:** 2
**Empirical Novelty And Significance:** 2
**Recommendation:** 3

**Clarity, Quality, Novelty And Reproducibility:**

**************Clarity**************

The paper is clearly written to understand overall method.

**************Quality**************

More experiments are required to validate the significance of proposed method. Also, it would be better to conduct more experiments to validate the contribution of each component.

**********Novelty**********

The proposed method has a limited novelty.

******************************Reproducibility******************************

The paper has fair reproducibility. The hyperparameters and experimental details are explained in the supplemnt.

**Strength And Weaknesses:**

**Strengths**

(1) This paper addresses the over-smoothing issue of graph neural networks, which is one of important topics to graph neural networks with deep layers.

(2) The proposed Deeper-GXX has demonstrated why stacking deep layer is important with the missing feature scenario and shown good performance in missing feature scenario.

**Weakness**

(1) I think that the novelty of this paper is limited.

- The decaying factor of proposed Weight-Decaying Graph Residual Connection seems very similar to $\alpha$ in APPNP [1].
- Also, Topology-guided Graph contrastive loss seems similar to self-supervised loss of SuperGAT [2] and PairNorm [3]. Specifically, the intuition of both papers is maximizing similarity between connected nodes and minimizing between unconnected nodes.

(2) Table 1 shows that Deeper-GXX shows good performance when the number of layer is 50. But, GCN shows better performance when the number of layer is lower. To validate the effectiveness of Deeper-GXX, I think comparison with baselines depending on the number of layers is needed.

(3) From Table 3, we can know that the weight decaying factor mostly contributes to the performance gain. Also, the performance gain of the similarity measure is marginal. It would be better to provide ablation studies on other dataset.

---

[1] Klicpera, Johannes, Aleksandar Bojchevski, and Stephan Günnemann. "Predict then propagate: Graph neural networks meet personalized pagerank." ICLR 2019.

[2] Kim, Dongkwan, and Alice Oh. "How to find your friendly neighborhood: Graph attention design with self-supervision." ICLR 2021.

[3] Zhao, Lingxiao, and Leman Akoglu. "Pairnorm: Tackling oversmoothing in gnns." ICLR 2020.

**Summary Of The Paper:**

The paper proposes Weight-Decaying Graph Residual Connection (WDG-ResNet) and Topology-Guided Graph Contrastive Loss (TGCL) to address shading neighbors and over-smoothing issues. Based on proposed WDG-ResNet and TGCL, the authors design an end-to-end model called Deeper-GXX to help GNNs perform well with deep layers.

**Summary Of The Review:**

Overall, I am leaning towards rejection. My major concern is the novelty and experimental results. If you address my concerns, I will raise my score.

=== post rebuttal ===

Thank you for your detailed response on the review. Even though some concerns are addressed, I still have concerns on the paper so that I maintain my score.

---

> ### Author Response · Authors · 2022-11-14
> **From authors: reply to concerns of reviewer 9Psf (1/2)**
>
> Thanks for your review and appreciation of our designed case study for the necessity of deeper GNN models. We would like to address your raised concerns as follows. Each concern raised by the reviewer is indexed by numbers, followed by our response.
>
> 1. **The decaying factor of the proposed WDG-ResNet is very similar to $\alpha$ in APPNP [Gasteiger et al., ICLR 2019], such that the novelty is limited**.
>
> We claim that they are different for the following reasons. According to JK-Net [Xu et al. ICML 2018] (as shown in Figure 3 of JK-Net paper), the $\alpha$ of APPNP [Gasteiger et al., ICLR 2019] (i.e., random walk with restart) is similar to the vanilla residual connections. The decaying factor of WDG-ResNet is inspired by the drawbacks that vanilla residual connections distort the non-Euclidean constraints of graph topology when adapted to GNNs, and they will cause our discovered sub-optimal “shading neighbor” effect. With this discovered effect, we propose the effective solution WDG-ResNet by leveraging the layer dependency to prevent the close neighbor information from being shaded. More importantly, we discover that the layer dependency has a close relationship with the optimal number of layers in deeper GNNs bridged by the hyperparameter $\lambda$, which is related to the diameter of the input graph. We show this discovery in Figure 5 and Table 6.
>
>
> 2. **Topology-guided graph contrastive loss (TGCL) seems similar to the self-supervised loss of SuperGAT [Kim et al., 2021 ICLR] and PairNorm [Zhao et al., 2020 ICLR], such that the novelty is limited**.
>
> For SuperGAT [Kim et al., ICLR 2021], it introduces a self-supervised loss for the vanilla GAT. As the reviewer mentioned, that self-supervised loss aims to maximize the similarity between connected nodes and minimize that between unconnected nodes. Here, we would like to emphasize that contrasting the existing edges and non-existing edges to guide self-supervised learning (or unsupervised learning) is a general principle and not solely attributed to SuperGAT. The authors of SuperGAT also claimed that “SuperGAT is the first study to analyze *self-supervised learning* on *graph attention* with *edge information*”. Based on this general contrasting principle, many pioneering works and follow-up works of SuperGAT have been proposed with their unique contributions, just like our Deeper-GXX. Therefore, we do not think the existence of SuperGAT hurts the novelty of Deeper-GXX.
>
> To support our above statement, we list related works other than SuperGAT that also utilize this general contrasting principle as follows: Eq.1 in TransE [Bordes et al., NeurIPS 2013],  Eq. 3 in node2vec [Grover et al., KDD 2016], Eq.1 in DGI [Velickovic et al., ICLR 2019], Eq.6 in GPT-GNN [Hu et al., KDD 2020], and Eq.3 in GraphCL [You et al., NeurIPS 2020]. Each of the listed works contributes its own thinking and adaptation. Similarly, in our TGCL loss of Deeper-GXX, we introduce the proximity measurement to weigh the closeness of node hidden representations (not proposed by SuperGAT), which is efficiently computed and proved to be effective (i.e., Experiments in Table 3 and Proof in Appendix B). Moreover, it is purely based on inherent information and does not need prior knowledge, which illustrates the difference of our TGCL from PairNorm [Zhao et al., ICLR 2020] as mentioned in your second concern.
>
> As for PairNorm [Zhao et al., ICLR 2020], it introduces a normalization layer that works to achieve the goal of keeping disconnected pair representations farther off. In the normalization layer, the hyperparameter $s$ takes the responsibility to scale the distance, i.e., Eq. 11 in PairNorm paper. We suppose how to get the optimal value for the hyperparameter $s$ requires the prior knowledge of the input graph, which may be hard to acquire in the real world scenario. Again, our TGCL does not require any prior knowledge, the distance indicator is purely from the input graph and easily obtained, i.e., Eq.6, and empirically performs well, i.e., Table 3 and Table 5.
>
> Besides, we would like to reorganize the paper and include the above analysis and include SuperGAT discussion in the camera ready.

---

> > ### Author Response · Authors · 2022-11-14
> > **From authors: reply to concerns of reviewer 9Psf (2/2)**
> >
> > 3. **GCN shows better performance when the number of layers is lower. To validate the effectiveness of Deeper-GXX, I think the comparison with baselines depending on the number of layers is needed**.
> >
> > We would like to point out that in Table 2, we report the best layer for each selected baseline, and the layer candidate pool is {2, 3, 4, 5, 6, 7, 8, 9, 10, 15, 20, 25, 30, 40, 50, 60}. As you mentioned, GCN indeed has better performance when the neural structure is shallow. For example, in Table 2, GCN’s optimal number of layers is 7 in the Cora dataset. However, please note that our Deeper-GXX could outperform the best shallow GCN across all datasets by ~4% through stacking layers effectively, especially when the input graph is imperfect, e.g., with missing features.
> >
> > 4. **The performance gain of the similarity measure is marginal. It would be better to provide ablation studies on other datasets**.
> >
> > As you suggested, we have expanded the ablation study in Table 3 by introducing datasets Citeseer, PubMed, Reddit, and OGB-Arxiv as follows, this additional ablation study is now attached in the updated paper as Table 5.
> >
> > Table. Ablation Study on CiteSeer, PubMed, Reddit, and OGB-Arxiv Data Sets w.r.t. Node
> > Classification Accuracy (following the same experimental setting as Table 3 in the paper)
> > ------- | Citeseer | Pubmed | Reddit | OGB-Arxiv
> > -------|--------|-------|--------|-------
> > Deeper-GXX  | 0.6655+/-0.0117   | 0.8185+/-0.0016 | 0.9721+/-0.0011 | 0.7401+/-0.0009
> > Deeper-GXX-S  | 0.6544+/-0.0099   | 0.8198+/-0.0012 | 0.9693+/-0.0036 | 0.7382+/-0.0008
> > Deeper-GXX-T  | 0.6567+/-0.0052   | 0.8150+/-0.0031 | 0.9654+/-0.0028 | 0.7363+/-0.0011
> > Deeper-GXX-D  | 0.5750+/-0.0244   | 0.8098+/-0.0047 | 0.9397+/-0.0042 | 0.7335+/-0.0024
> >
> > We can observe from the table that the original WDG-ResNet (Deeper-GXX) still outperforms the simplified WDG-ResNet (Deeper-GXX-S) in most cases, e.g., with 1% testing accuracy more than Simplified WDG-ResNet on Citeseer). We would like to point out that 0.5-1% is not a very small improvement: similar self-improvements can be found in GCNII [Chen et al., ICML 2020], PairNorm [Zhao et al., ICLR 2020], DropEdge [Rong et al., ICLR 2020], etc. Especially, in the large dataset OGB-Arxiv whose testing size is 48,603, 0.2% indicates numerous additional nodes (in the order of 10^2) that have been correctly classified. Moreover, the reason we propose an alternative to WDG-ResNet is to achieve comparable performance as Deeper-GXX, but with much better efficiency. The performance gap between Deeper-GXX-S and Deeper-GXX is expected to be small (say 0.2%-0.3%) as shown in Table 6 (e.g., Reddit, OGB-Arxiv, etc.). Therefore, we can summarize that (1) if performance is the top priority, one should choose the original algorithm, and (2) if efficiency is most important, one should choose the simplified version.
> >
> > Reference
> >
> > [Xu et al. ICML 2018] Keyulu Xu, Chengtao Li, Yonglong Tian, Tomohiro Sonobe, Ken-ichi Kawarabayashi, Stefanie Jegelka: Representation Learning on Graphs with Jumping Knowledge Networks. ICML 2018
> >
> > [Gasteiger et al., ICLR 2019] Johannes Klicpera, Aleksandar Bojchevski, Stephan Günnemann: Predict then Propagate: Graph Neural Networks meet Personalized PageRank. ICLR 2019
> >
> > [Kim et al., ICLR 2021] Dongkwan Kim, Alice Oh: How to Find Your Friendly Neighborhood: Graph Attention Design with Self-Supervision. ICLR 2021
> >
> > [Bordes et al., NeurIPS 2013] Antoine Bordes, Nicolas Usunier, Alberto García-Durán, Jason Weston, Oksana Yakhnenko: Translating Embeddings for Modeling Multi-relational Data. NeurIPS 2013
> >
> > [Grover et al., KDD 2016] Aditya Grover, Jure Leskovec: node2vec: Scalable Feature Learning for Networks. KDD 2016
> >
> > [Velickovic et al., ICLR 2019] Petar Velickovic, William Fedus, William L. Hamilton, Pietro Liò, Yoshua Bengio, R. Devon Hjelm: Deep Graph Infomax. ICLR 2019
> >
> > [Hu et al., KDD 2020] Ziniu Hu, Yuxiao Dong, Kuansan Wang, Kai-Wei Chang, Yizhou Sun: GPT-GNN: Generative Pre-Training of Graph Neural Networks. KDD 2020
> >
> > [You et al., NeurIPS 2020] Yuning You, Tianlong Chen, Yongduo Sui, Ting Chen, Zhangyang Wang, Yang Shen: Graph Contrastive Learning with Augmentations. NeurIPS 2020
> >
> > [Zhao et al., ICLR 2020] Lingxiao Zhao, Leman Akoglu: PairNorm: Tackling Oversmoothing in GNNs. ICLR 2020

---

> ### Author Response · Authors · 2022-11-18
> **From authors: discussions are welcome**
>
> Dear Reviewer,
>
> Until now we have not received your discussion and the time will be up soon, would you please discuss based on what we replied?
>
> Best,
>
> Authors

---

> > ### Author Response · Authors · 2022-12-09
> > **From authors: discussions are still welcome**
> >
> > Dear Reviewer,
> >
> > The second discussion phase seems to come to an end very soon, we still did not know how you evaluate our response answer for your raised concerns. We sincerely thank you for your time and hard work in advance!
> >
> > Best,
> >
> > Authors

---

### Official Review · Reviewer_y1R7 · 2022-10-31

**Confidence:** 3
**Correctness:** 3
**Technical Novelty And Significance:** 3
**Empirical Novelty And Significance:** 3
**Recommendation:** 5

**Clarity, Quality, Novelty And Reproducibility:**

Clarity: The work is generally clear and easy to understand, apart from the parts pertaining to the `sim` part of TGCL. This work does have a sizeable number of small writing errors, but these should be easy to fix (I listed a bunch in my "Nitpicks" section).

Quality: Experiments in this work are relatively thorough, but (as I mentioned in the "Weaknesses" section), some more comparisons could be useful, and motivation for the `sim` part of the loss is lacking.

Reproducibility: The work provides a good number of implementation details, and thus seems reproducible.

**Strength And Weaknesses:**

=== Strengths ===

(S1): This work explores generalizable modifications to GNNs that could be applicable to a broad range of architectures/problems.

(S2): The authors define the "shading neigbhors effect", which they use to motivate their approach better.



=== Weaknesses ===

(W1): The motivation behind the `sim` component of the architecture (Equation 4) is unclear to me, and the gains from adding it border on statistical significance:

- The authors mention that this component "expands the hypothesis space of GNNs", but I'm not sure how to understand that. What are we trying to achieve with this term?

- Are we backpropagating through `sim` during training?

- Judging from results in Table 1 and 2, the gain from adding this term is minimal, bordering on statistical significance. I agree there is *some* signal if one integrates the small increments across all the different experiments and setups, but (as the authors note), this component also increases compute requirements and complicates things conceptually.

(W2): Some more comparisons would help to contextualize the results:

- Simplified WDG could be compared to an ablation where the weights are learned scalars, instead of being fixed to `e^{-l/lambda}`. This would show whether the decaying-by-design weighing is indeed what is causing the improvements, or the sheer fact that there is a post-activation coefficient smaller than 1 is already enough.

- The authors compare against ResNet-style GNNs, but there's another approach (also extremely practical and common), which is to concatenate the hidden representations across all layers, project that concatenated vector to a suitable dimension, and use that for the downstream task. This is sometimes referred to as "Jumping Knowledge", and obtained by setting `jk="cat"` in `pytorch_geometric`.



=== Nitpicks ===

Here I include some final nitpicks, which did not affect my score; they are here just to help improve the paper.

- "user’s immediate neighbor information may not reflect his/her categorical information": use "their" instead of "his/her"

- "information collected from faraway neighbors become dominant": "become" -> "becomes"

- "This radioactive decay": drop the word "radioactive"

- "simpler version of Eq. 4 by fixing the sim": I think "fixing" should rather be "removing"?

- Missing space before citation in Section 2.3

- I'd call Section 3 "Experiments" (plural) instead of "Experiment"

- "could be found in Appendix" (twice): I would say "can" instead

- "shallow GCN with ResNet has enough capability": "capability" -> "capacity"

- "we need more neighbors aggregated for collecting their partial features to compensate for the missing such that stacking GNN layers works": this sentence seems broken, please rephrase

- "conduct the ablation study": "the" -> "an"


**Summary Of The Paper:**

This paper proposes two tricks to improve GNN performance at large depths: WDG-ResNet (weighted residual connections) and TGCL (additional loss term). Through experiments and ablations they show that a GNN enhanced with both modifications is better-behaved when scaling up depth.

**Summary Of The Review:**

The premise of this work is overall sound, and the results look promising, but some extra intuition is needed, and I would welcome some more baseline comparisons. For my initial evaluation I lean slightly to the negative side, but I'm open to discussing with the authors and other reviewers.

---

> ### Author Response · Authors · 2022-11-14
> **From authors: reply to concerns of reviewer y1R7 (1/2)**
>
> Thanks so much for your review and commending our discovery of the “shading neighbor” effect, as well as the suggestions for improving our paper's readability. We will address each raised concern as follows. Each concern raised by the reviewer is indexed by numbers, followed by our response.
>
> 1. **How to understand “expands the hypothesis space of GNNs” by the $sim(\cdot)$ component in Eq.4**?
>
> From the viewpoint that neural networks simulate certain functions, a specific neural network provides a hypothesis space where the function can be generated (i.e., trained through samples and losses) as shown on page 14 of [this lecture](https://davidrosenberg.github.io/mlcourse/Archive/2017/Lectures/14b.neural-networks.pdf). Compared with a constant similarity weight like 1 in ResNet, $sim(\cdot)$ in WDG-ResNet is not instanced by a certain constant but a learnable similarity weight by the hidden representations, which potential is used for fitting the training samples, such that the hypothesis of GNNs expands.
>
> 2. **Are we backpropagating through $sim(\cdot)$ during training**?
>
> If we understand your question correctly, are you asking whether $H^{(1)}$ and $H^{(l)}$ inside the $sim(\cdot)$ function are involved in backpropagation? If so, the answer is yes. Because we do not add the stop gradient operation in $sim(\cdot)$, and $sim(H^{(1)},H^{(l)})$ is defined as $sim(H^{(1)},H^{(l)})=exp(H^{(1)} H^{(l){\top}} / (|H^{(1)}||H^{(l)}|))$, where the layer-wise hidden representations $H^{(1)}$ and $H^{(l)}$ participate in backpropagation.
>
> 3. **The performance gain of the learnable similarity $sim(\cdot)$ between layers is marginal, but the computational complexity may be costly**.
>
> For your raised concern, in the original paper, we propose Simplified WDG-ResNet, which replaces the $sim(\cdot)$ with the constant to save efficiency. The reason we propose an alternative to WDG-ResNet is to achieve comparable performance as Deeper-GXX, but with much better efficiency.
>
> To investigate the exact performance of WDG-ResNet (instanced in Deeper-GXX) and Simplified WDG-ResNet (instanced in Deeper-GXX-S), we conduct the ablation studies in Table 3 in the paper. Additionally, the ablation study in Table 3 is further expanded as follows by including the results on additional datasets, Citeseer, PubMed, Reddit, and OGB-Arxiv. The following table is now attached in the updated paper as Table 5. In the following table, we can observe that the performance gap between Deeper-GXX-S and Deeper-GXX is small as expected, say 0.2%-0.3% as shown in the following table (e.g., Reddit, OGB-Arxiv, etc.). Deeper-GXX-S satisfies our expectation that the approximated method could achieve competitive performance but with computational complexity saved.
>
> Table. Ablation Study on CiteSeer, PubMed, Reddit, and OGB-Arxiv Data Sets w.r.t. Node
> Classification Accuracy (following the same experimental setting as Table 3 in the paper)
> ------- | Citeseer | Pubmed | Reddit | OGB-Arxiv
> -------|--------|-------|--------|-------
> Deeper-GXX  | 0.6655+/-0.0117   | 0.8185+/-0.0016 | 0.9721+/-0.0011 | 0.7401+/-0.0009
> Deeper-GXX-S  | 0.6544+/-0.0099   | 0.8198+/-0.0012 | 0.9693+/-0.0036 | 0.7382+/-0.0008
> Deeper-GXX-T  | 0.6567+/-0.0052   | 0.8150+/-0.0031 | 0.9654+/-0.0028 | 0.7363+/-0.0011
> Deeper-GXX-D  | 0.5750+/-0.0244   | 0.8098+/-0.0047 | 0.9397+/-0.0042 | 0.7335+/-0.0024
>
> Also, we would like to mention that, the original WDG-ResNet still performs the best, e.g., with 1% testing accuracy more than Simplified WDG-ResNet on Citeseer). We would like to point out that 0.5%-1% is not a very small improvement: similar self-improvements can be found in GCNII [Chen et al., ICML 2020], PairNorm [Zhao et al., ICLR 2020], and DropEdge [Rong et al., ICLR 2020]. Especially, in the large dataset OGB-Arxiv whose testing size is 48603, 0.2% indicates numerous additional nodes (in the order of 10^2) that have been correctly classified.
>
> Finally, we can summarize that (1) **if performance is the top priority, one should choose the original algorithm**, and (2) **if efficiency is most important, one should choose the simplified version**.

---

> > ### Author Response · Authors · 2022-11-14
> > **From authors: reply to concerns of reviewer y1R7 (2/2)**
> >
> > 4. **Simplified WDG should be compared to an ablation where the weights are learned scalars, instead of being fixed to $e^{-l/\lambda}$**.
> >
> > The “learnable weights without decaying” you suggested is exactly in our ablation study, i.e., Deeper-GXX-D in Table 3, which removes the decaying factor but maintains the learnable weights. It can be found in the ablation study in Section 3.4 - Ablation Study and the corresponding performance is shown in Table 3. Also, the ablation study in Table 3 is comprehensive by including Deeper-GXX-D, Deeper-GXX-S, and Deeper-GXX-T. The “$sim(\cdot)$ being fixed as $e^{-l/\lambda}$” part you mentioned is our Deeper-GXX-S, which replaces the learnable weights with the constant but remains the decaying factor. Moreover, we have Deeper-GXX-T standing for the removal of the contrastive loss TGCL in Deeper-GXX. Also, as discussed above, the additional ablation study executed on more datasets (i.e., CiteSeer, PubMed, Reddit, and OGB-Arxiv) is now available and attached in the updated paper as Table 5.
> >
> > 5. **Weak baselines for not including JK-Net [Xu et al. ICML 2018]**.
> >
> > We would like to clarify that our selected baselines are SOTA. Please refer to **the second bullet point** of the top section titled “From authors: reply to shared concerns from different reviewers”. Briefly, the suggested baseline JK-Net [Xu et al. ICML 2018] has been outperformed by our selected baseline GCNII as shown in [Chen et al., ICML 2020].
> >
> > Thanks again for your suggested **nitpicks**, all of them are addressed in the updated version.
> >
> > Reference
> >
> > [Zhao et al., ICLR 2020] Lingxiao Zhao, Leman Akoglu: PairNorm: Tackling Oversmoothing in GNNs. ICLR 2020
> >
> > [Chen et al., ICML 2020] Ming Chen, Zhewei Wei, Zengfeng Huang, Bolin Ding, Yaliang Li: Simple and Deep Graph Convolutional Networks. ICML 2020
> >
> > [Rong et al., ICLR 2020] Yu Rong, Wenbing Huang, Tingyang Xu, Junzhou Huang: DropEdge: Towards Deep Graph Convolutional Networks on Node Classification. ICLR 2020

---

> > > ### Comment · Reviewer_y1R7 · 2022-11-27
> > > **Response to authors**
> > >
> > > Thanks for a detailed response!
> > >
> > > Regarding JK-Nets, I wasn't really asking for a comparison with the _specific_ model proposed in the JK-Nets paper (which includes all the architectural/hyparparameter choices the authors made at the time). As you said, that was since surpassed by other approaches, but I think it's not necessarily the case that the `jk="cat"` trick is worse than e.g. skip connections, a variety of factors could be at play. I was thinking of `jk="cat"` as being a generic modification one can do to pretty much any GNN. As you say, for each baseline you added the residual connection if it helped, treating it as a binary hyperparameter; I would consider adding another hyperparameter saying whether to enable `jk="cat"`. Does that make sense?
> > >
> > > (By the way: I think despite what the original timeline said, you should still be able to respond to this comment)

---

> > > > ### Author Response · Authors · 2022-12-01
> > > > **Response to reviewer y1R7**
> > > >
> > > > Thanks so much for your reply! Based on your comments, we regard the jumping knowledge (i.e., JK) as a general technique with the same role of residual connection, and then replace the residual connection with it to see whether the performance will be improved or not.
> > > >
> > > > To be specific, we did the new experiments as follows. In our Deeper-GXX, we use JK as a substitute for the proposed weight-decaying residual connection (i.e., WDG), which model is denoted as **Deeper-GXX(-WDG +JK)**. The experimental results are shown as follows.
> > > >
> > > > ------- | Cora | Citeseer | Pubmed | Reddit | OGB-Arxiv
> > > > -------|--------|-------|--------|-------|-------
> > > > Deeper-GXX(-WDG +JK)  | 0.7955+/-0.0078   | 0.6600+/-0.0085 | 0.8161+/-0.0038 | 0.9659+/-0.0046 | 0.7368+/-0.0012
> > > > Deeper-GXX  | **0.8059+/-0.0028**   | **0.6655+/-0.0117** | **0.8185+/-0.0016** | **0.9721+/-0.0011** | **0.7401+/-0.0009**
> > > >
> > > > In the above table, the number of layers is set to be 60 for the four relatively small benchmark datasets, and the number of layers is 20 for the large and heterophyllous OGB dataset. We observe that our vanilla Deeper-GXX still outperforms Deeper-GXX(-WDG +JK) in all datasets by 0.5~1%. **More importantly**, the computational cost of Deeper-GXX(-WDG +JK) is much more expensive than Deeper-GXX because of the concatenation of features to realize deeper models. It means that, when the number of layers increases, the number of parameters for stacking one signal layer increases cumulatively. The detailed analysis is stated below.
> > > >
> > > > **Theoretically**, in jumping knowledge (i.e., JK), the dimension of the $n$-th hidden layer can be computed by $d_{in}^{(n)} = \sum_{i=1}^{n-1}d_{out}^{(i)}$, where $d_{in}^{(i)}$ is the dimension of the input feature and  $d_{out}^{(i)}$ is the dimension of the output feature at the $i$-th layer. **Empirically**, to represent this complexity with the specific number of parameters, for the above performance (i.e., in the OGB dataset, for 20 layers), our Deeper-GXX only needs 301040 parameters, **but Deeper-GXX(-WDG +JK) needs around 3x as many parameters**, i.e., 891040.
> > > >
> > > > The overall conclusion is that using JK could perform well (even though still falls behind ours), but it tends to have a more expensive computational cost. And we will add the above analysis in the future camera ready.
> > > >
> > > > Thanks again for your comments! We sincerely appreciate your time and would like to discuss with you for any other future concerns.

---

> ### Author Response · Authors · 2022-11-18
> **From authors: discussions are welcome**
>
> Dear Reviewer,
>
> Until now we have not received your discussion and the time will be up soon, would you please discuss based on what we replied?
>
> Best,
>
> Authors

---

### Official Review · Reviewer_YeUe · 2022-11-05

**Confidence:** 4
**Correctness:** 3
**Technical Novelty And Significance:** 2
**Empirical Novelty And Significance:** Not applicable
**Recommendation:** 5

**Clarity, Quality, Novelty And Reproducibility:**

The paper is overall clarity written and easy to follow, all the equations, figures and tables are clearly described.
The hyperparameter tuning is provided in the appendix and I believe the reproducibility of this paper is good.
However, the novelty is limited as I have mentioned above, especially in the simplified WDG-ResNet.

**Strength And Weaknesses:**

**Strength**
- The paper is overall well written and easy to follow. The figures and tables are well described.
- The finding that "optimal decaying factor is close to the diameter of input graphs (if it is connected) or the largest component (if it does not have many separate components)." is interesting and may inspire some follow up works.
- The reproducibility of this paper looks good to me.

**Weakness**
- The technical contribution is limited. Concerning the two tricks, the first trick has been widely studied in the literature, such as the Jumping Knowledge Networks[1] or the Predict then Propagate [2], the idea is quite similar that by incoporating information from the longer distance nodes while avoiding the domination of them, which is a trade-off between close and distant nodes. This paper propose a decay funnction which is a degraded version of above methods. The second trick is widely studied in contrastive learning that adding weights on the positive and negative pairs by the hardness of sample, where the distance is utilized as hardness here.
- The critical design is the  learnable similarity between the layers, which distinguish this paper from the ResNet. However, due to the complexity and training efficiency proble, the simiplified WDG-ResNet ignores the learnable parameters and this makes the proposed WDG-ResNet quite similar to the ResNet with only layer-wise constant decays. This is quite weired and the motivation of WDG-ResNet becomes unclear.
- The experiments are conducted on the four datasets: Cora, Citeseer, Pubmed and Reddit, which is quite small. Furthermore, these datasets are representative ones with high homophily, where neighborhood nodes tends to share similar labels. The performance of the proposed method on large datasets and lower homophily datasets is not convincing.
- The baselines are weak, many GNN methods such as [1][2] and graph contrastive learning methods are missed.


[1] Representation Learning on Graphs with Jumping Knowledge Networks
[2] Predict then Propagate: Graph Neural Networks meet Personalized PageRank

**Summary Of The Paper:**

This paper studies the problem of vanishing gradient and over-smoothing in graph neural networks which is highly related to the deep GNN training. This paper proposes two tricks into existing GNN layers named: Weight-Decaying Graph Residual Connection module (WDG-ResNet) and Topology-Guided Graph Contrastive Loss (TGCL). Experiments conducted on four datasets demonstrate the effectiveness of the proposed method.

**Summary Of The Review:**

Although the problem studied in the paper is intersting and the paper is well written, my major concerns are the technical contribution (both two tricks) and the experimental results (datasets and baselines). The authors are expected to provide detailed evidences to address the above concerns for increasing the final scores.

---

> ### Author Response · Authors · 2022-11-14
> **From authors: reply to concerns of reviewer YeUe (1/2)**
>
> Thanks for your review, and we are very glad to learn that you commend our discovery in the optimal layer dependency of WDG-ResNet on the diameter of the input graph, which could provide research potential for the research community. Next, we would like to address your raised concerns as follows. Each concern raised by the reviewer is indexed by numbers, followed by our response.
>
> 1. **Our technical contribution is limited because of the existence of JK-Net [Xu et al. ICML 2018] and APPNP [Gasteiger et al., ICLR 2019]**.
>
> Please refer to **the third bullet point** of the top section titled “From authors: reply to shared concerns from different reviewers”.
>
> Also, the reviewer mentioned that contrastive learning is widely used in graph problems, which also jeopardizes our contribution. Indeed, the general idea of contrasting is a popular solution to address the over-smoothing problem in deeper GNNs, and many following works depend on that but develop various improvements for effectiveness and efficiency. PairNorm [Zhao et al., ICLR 2020] introduces the normalization layer controlled by a constant to scale the distance between pair node hidden representations, and DGN [Zhou et al., NeurIPS 2020] introduces the mutual information constraint to a grouping (i.e., similarity measurement) of nodes. (Note that these two are also baselines in our paper.). For a similar rationale, we develop the simple but effective TGCL contrastive loss in Deeper-GXX. It avoids the hard-to-acquire prior contrastive knowledge like constant hyperparameter or grouping methods, but transfers them into the easy-to-obtain information, i.e., adjacency vector associated with the proper distance function. The theoretical effectiveness analysis is provided in Appendix B, and the empirical performance is shown in the ablation study in Table 3, where Deeper-GXX-T stands for the removal of TGCL from Deeper-GXX.
>
> 2. **Motivation for the learnable similarity between layers in WDG-ResNet is not clear, because the simplified WDG-ResNet achieves competitive performance**.
>
> Both WDG-ResNet (instanced in Deeper-GXX) and Simplified WDG-ResNet (instanced in Deeper-GXX-S) are our originally proposed methods. In particular, the proposed Simplified WDG-ResNet is inspired by the original WDG-ResNet.
>
> We also would like to emphasize that Simplified WDG-ResNet (i.e., Deeper-GXX-S) achieves comparable performance but with much better efficiency than WDG-ResNet (i.e., Deeper-GXX). In the meanwhile, the original WDG-ResNet still performs the best, e.g., with 1% testing accuracy more than Simplified WDG-ResNet on Citeseer shown in Table 5). We would like to point out that 0.5-1% is not a very small improvement: similar self-improvements can be found in GCNII [Chen et al., ICML 2020], PairNorm [Zhao et al., ICLR 2020], DropEdge [Rong et al., ICLR 2020], etc. Especially, in the large dataset OGB-Arxiv whose testing size is 48,603, 0.2% indicates numerous additional nodes (in the order of 10^2) that have been correctly classified. Moreover, the reason we propose an alternative to WDG-ResNet is to achieve comparable performance as Deeper-GXX, but with much better efficiency. The performance gap between Deeper-GXX-S and Deeper-GXX is expected to be small (say 0.2%~0.3%) as shown in Table 6 (e.g., Reddit, OGB-Arxiv, etc.).
>
> As suggested by the reviewer, we extend the ablation studies to extra datasets for comprehensiveness. The additional experiments for expanding Table 3 in the paper are shown as follows. This result is now attached to the paper as Table 6.
>
> Table. Ablation Study on CiteSeer, PubMed, Reddit, and OGB-Arxiv Data Sets w.r.t. Node
> Classification Accuracy (following the same experimental setting as Table 3 in the paper)
> ------- | Citeseer | Pubmed | Reddit | OGB-Arxiv
> -------|--------|-------|--------|-------
> Deeper-GXX  | 0.6655+/-0.0117   | 0.8185+/-0.0016 | 0.9721+/-0.0011 | 0.7401+/-0.0009
> Deeper-GXX-S  | 0.6544+/-0.0099   | 0.8198+/-0.0012 | 0.9693+/-0.0036 | 0.7382+/-0.0008
> Deeper-GXX-T  | 0.6567+/-0.0052   | 0.8150+/-0.0031 | 0.9654+/-0.0028 | 0.7363+/-0.0011
> Deeper-GXX-D  | 0.5750+/-0.0244   | 0.8098+/-0.0047 | 0.9397+/-0.0042 | 0.7335+/-0.0024
>
> Finally, for this concern, we can summarize that (1) if performance is the top priority, one should choose the original algorithm, and (2) if efficiency is most important, one should choose the simplified version.

---

> > ### Author Response · Authors · 2022-11-14
> > **From authors: reply to concerns of reviewer YeUe (2/2)**
> >
> > 3. **Only used four small homophilous datasets**.
> >
> > We would like to clarify that we used large heterophyllous datasets in the original paper. Please refer to **the first bullet point** of the top section titled “From authors: reply to shared concerns from different reviewers”.
> >
> > 4. **The baselines are weak for not including JK-Net [Xu et al. ICML 2018] and APPNP [Gasteiger et al., ICLR 2019]**.
> >
> > We would like to clarify that our selected baselines are SOTA. Please refer to **the second bullet point** of the top section titled “From authors: reply to shared concerns from different reviewers”. Briefly, the suggested baselines JK-Net [Xu et al. ICML 2018] and APPNP [Gasteiger et al., ICLR 2019] have been outperformed by our selected baseline GCNII, as shown in [Chen et al., ICML 2020].
> >
> > Reference
> >
> > [Xu et al. ICML 2018] Keyulu Xu, Chengtao Li, Yonglong Tian, Tomohiro Sonobe, Ken-ichi Kawarabayashi, Stefanie Jegelka: Representation Learning on Graphs with Jumping Knowledge Networks. ICML 2018
> >
> > [Gasteiger et al., ICLR 2019] Johannes Klicpera, Aleksandar Bojchevski, Stephan Günnemann: Predict then Propagate: Graph Neural Networks meet Personalized PageRank. ICLR 2019
> >
> > [Rong et al., ICLR 2020] Yu Rong, Wenbing Huang, Tingyang Xu, Junzhou Huang: DropEdge: Towards Deep Graph Convolutional Networks on Node Classification. ICLR 2020
> >
> > [Zhao et al., ICLR 2020] Lingxiao Zhao, Leman Akoglu: PairNorm: Tackling Oversmoothing in GNNs. ICLR 2020
> >
> > [Chen et al., ICML 2020] Ming Chen, Zhewei Wei, Zengfeng Huang, Bolin Ding, Yaliang Li: Simple and Deep Graph Convolutional Networks. ICML 2020
> >
> > [Zhou et al., NeurIPS 2020] Kaixiong Zhou, Xiao Huang, Yuening Li, Daochen Zha, Rui Chen, Xia Hu: Towards Deeper Graph Neural Networks with Differentiable Group Normalization. NeurIPS 2020

---

> ### Author Response · Authors · 2022-11-18
> **From authors: discussions are welcome**
>
> Dear Reviewer,
>
> Until now we have not received your discussion and the time will be up soon, would you please discuss based on what we replied?
>
> Best,
>
> Authors

---

> > ### Author Response · Authors · 2022-12-09
> > **From authors: discussions are still welcome**
> >
> > Dear Reviewer,
> >
> > The second discussion phase seems to come to an end very soon, we still did not know how you evaluate our response answer for your raised concerns. We sincerely thank you for your time and hard work in advance!
> >
> > Best,
> >
> > Authors

---

### Author Response · Authors · 2022-11-14
**From authors: reply to shared concerns of different reviewers (1/2)**

Thanks for all the reviewers’ time and hard work. There are several shared misunderstandings from multiple reviewers that became the major rejection reasons. So we would like to clarify them here.

*Before the clarification, we would like to mention the manuscript is updated by adding additional ablation experiments suggested by reviewers in Table 5 in Appendix A.3. Therefore, some index numbers of tables may be affected and reordered. All the index numbers mentioned below are referring to the updated manuscript*.

1. **Only utilized four small homophilous datasets in the paper**, which is shared by reviewer YeUe and reviewer G8Tr.

The reviewers YeUe and G8Tr are concerned that we only use the small homophilous datasets in the paper, and reviewer G8Tr also recommends the OGB dataset. We would like to clarify that we have already used the OGB dataset and the corresponding results have already been included in the initial submission. To be specific, we utilized the large dataset OGB-Arxiv with 169,343 nodes and 1,166,243 edges. This is also a heterophyllous dataset, whose edge homophily is only 0.222 measured by [Lim et al., GLB@WWW 2021]. In the OGB-Arxiv dataset, we compared all selected baselines with our proposed Deeper-GXX, and our Deeper-GXX also outperforms these baselines. Please refer to the paper Section 3.1 Experiment Setup - Datasets, and performance comparison in Figures 3a and 3b.

2. **Weak baselines in the paper because of not including JK-Net [Xu et al. ICML 2018] and APPNP [Gasteiger et al., ICLR 2019]**, which is shared by reviewer YeUe and reviewer y1R7.

We would like to clarify that the selected deeper baselines are state-of-the-art. JK-Net [Xu et al. ICML 2018] and APPNP [Gasteiger et al., ICLR 2019] have been outperformed by GCNII as reported in [Chen et al., ICML 2020]. For our deeper GNN baselines, we selected various kinds of techniques, i.e., GCNII [Chen et al., ICML 2020] to represent the layer aggregation techniques, PairNorm [Zhao et al., ICLR 2020] and DGN [Zhou et al., NeurIPS 2020] for the contrastive techniques (i.e., distance function in PairNorm, mutual information in DGN), and DropEdge [Rong et al., ICLR 2020] for ensemble learning techniques. Also, we equip each baseline with the residual connection layers if the residual can improve its performance.

3. **The limited technical contribution or novelty of our Deeper-GXX due to the existence of JK-Net [Xu et al. ICML 2018] and/or APPNP [Gasteiger et al., ICLR 2019]**, which is shared by reviewer YeUe and reviewer 9Psf.

Reviewer YeUe and 9Psf mentioned that JK-Net or APPNP aims to refine the layer aggregation mechanism to effectively obtain the longer-distance neighbor information, which seems like what our Deeper-GXX does. We would like to point out that, how to re-design the aggregation mechanism (i.e., aggregating layers or replacing layers with the proper stationary distribution) and how to obtain the far-away neighbor information to deepen GNNs is the general theme in the research direction of deep graph models. Besides JK-Net [Xu et al., ICML 2018] and APPNP [Gasteiger et al., ICLR 2019], the following works GCNII [Chen et al., ICML 2020], PPRGo [Bojchevski et al., KDD 2020], and GPR-GNN [Chien et al., ICLR 2021] all fall into this category, and their contributions are to address previous limitations by manipulating layers. The same theme applies to our Deeper-GXX. First, we find that simply adding residual connections on GNNs ignores the non-euclidean constraints of graph structures and causes the “shading neighbor” effect, which means the close neighbor’s information becomes less and less important as layers stack. We have theoretically (Eq. 3) and empirically (Appendix A.4) shown this effect.

To the best of our knowledge, this shading effect has not been discovered before. In the early work JK-Net [Xu et al., 2018 ICML], the authors visualized that the vanilla residual connection on GNNs is similar to a random walk with restart on graphs (i.e., APPNP [Gasteiger et al., ICLR 2019]). In the following deeper GNNs works, PairNorm [Zhao et al., ICLR 2020], DropEdge [Rong et al., ICLR 2020], and GCNII [Chen et al., ICML 2020] incorporate the vanilla (or similar) residual connection into their models and leave the “shading neighbor” effect untouched. With this newly discovered effect, we propose an effective solution named WDG-ResNet by leveraging the layer dependency to prevent the close neighbor information from being shaded. More importantly, we discover that the layer dependency has a close relationship with the optimal number of layers in deeper GNNs, which is bridged by the hyperparameter $\lambda$. We also discover the optimal $\lambda$ heavily depends on the diameter of the input graph, which tends to reduce the search complexity for hyperparameter tuning in deeper GNNs. And we show this discovery in Figure 5 and Table 6.

---

> ### Author Response · Authors · 2022-11-14
> **From authors: reply to shared concerns of different reviewers (2/2)**
>
> Based on the above reasons, we sincerely hope the reviewers could check the paper again for these contributions that set it apart from state-of-the-art.
>
>
> Reference
>
> [Xu et al. ICML 2018] Keyulu Xu, Chengtao Li, Yonglong Tian, Tomohiro Sonobe, Ken-ichi Kawarabayashi, Stefanie Jegelka: Representation Learning on Graphs with Jumping Knowledge Networks. ICML 2018
>
> [Gasteiger et al., ICLR 2019] Johannes Klicpera, Aleksandar Bojchevski, Stephan Günnemann: Predict then Propagate: Graph Neural Networks meet Personalized PageRank. ICLR 2019
>
> [Rong et al., ICLR 2020] Yu Rong, Wenbing Huang, Tingyang Xu, Junzhou Huang: DropEdge: Towards Deep Graph Convolutional Networks on Node Classification. ICLR 2020
>
> [Zhao et al., ICLR 2020] Lingxiao Zhao, Leman Akoglu: PairNorm: Tackling Oversmoothing in GNNs. ICLR 2020
>
> [Chen et al., ICML 2020] Ming Chen, Zhewei Wei, Zengfeng Huang, Bolin Ding, Yaliang Li: Simple and Deep Graph Convolutional Networks. ICML 2020
>
> [Bojchevski et al., KDD 2020] Aleksandar Bojchevski, Johannes Klicpera, Bryan Perozzi, Amol Kapoor, Martin Blais, Benedek Rózemberczki, Michal Lukasik, Stephan Günnemann: Scaling Graph Neural Networks with Approximate PageRank. KDD 2020
>
> [Zhou et al., NeurIPS 2020] Kaixiong Zhou, Xiao Huang, Yuening Li, Daochen Zha, Rui Chen, Xia Hu: Towards Deeper Graph Neural Networks with Differentiable Group Normalization. NeurIPS 2020
>
> [Lim et al., GLB@WWW 2021] Derek Lim, Xiuyu Li, Felix Hohne, Ser-Nam Lim: New Benchmarks for Learning on Non-Homophilous Graphs. GLB@WWW 2021
>
> [Chien et al., ICLR 2021] Eli Chien, Jianhao Peng, Pan Li, Olgica Milenkovic: Adaptive Universal Generalized PageRank Graph Neural Network. ICLR 2021

---

### Decision · Program_Chairs · 2023-01-20

**Decision:**

Reject

**Justification For Why Not Higher Score:**

Novelty is limited due to similarity to existing techniques, and some other issues with experimental design / motivation of learnable similarity component.

**Justification For Why Not Lower Score:**

N/A

**Metareview: Summary, Strengths And Weaknesses:**

The paper proposes two modifications for GNNs to allow deeper models, via a decay parameter applied to residual connections (WDG), and a contrastive loss approach (TGCL). They also propose a "shading neighbors effect" to motivate their decay-based approach, including an interesting finding that the optimal decay factor is related to the diameter of input graphs (or its largest component).

In general, reviewers find the paper clear and easy to follow, and appreciate the finding on the optimal decay factor and graph diameter. However, there are a number of key issues pointed out by the reviewers, particularly:

- Novelty: reviewers found that the novelty is relatively limited, considering similarity to existing techniques, such as APPNP for their decay component (WDG), and SuperGAT for their contrastive loss component (TGCL). In the rebuttal, the author response states that $\alpha$ is comparable to vanilla residual connections, but it seems reasonable to me to consider $\alpha$ as decay parameter in APPNP. For SuperGAT, the author response states that many other papers also use contrastive approaches, but this still leaves the overall issue that TGCL is similar to existing techniques with relatively limited technical contribution.

- Experimental design (e.g. reviewers noted that experiments in table 1 are done with 60 hidden layers, which is suboptimal for some baselines (as the authors agreed). The authors note that other experiments (table 2) select the best number of layers (which is reasonable, though they are for a different experiment involving attribute masking). Reviewers also found the number or variety of datasets (e.g. in terms of homophily / heterophily) limited. Other issues include the motivation and performance benefit of the learnable similarity ("sim") module being lacking (this observation still remains in the additional ablation results provided). The author responses did help in addressing some reviewers concerns, and I thank the authors for their efforts in addressing issues and improving the paper.

In the end, reviewers and AC agree that while the work has merits, due to these issues, the work is not yet ready for publication at ICLR. The reviews offer a number of helpful suggestions for improvement, so I encourage the authors to continue improving the paper based on the reviews for future submission.